# AHa-Bench: Benchmarking Audio Hallucinations in Large Audio-Language Models

## Is That an AHa? Or Just a Hallucination?

**Xize Cheng**[1,2*]  **Dongjie Fu**[1*]  **Chenyuhao Wen**[1*]  **Shannon Yu**[3]  **Zehan Wang**[1]
**Shengpeng Ji**[1]  **Siddhant Arora** [4]  **Tao Jin**[1]  **Shinji Watanabe**[4]  **Zhou Zhao**[1,2†]
Zhejiang University[1]  Shanghai Artificial Intelligence Laboratory[2]
Independent Researcher[3] Carnegie Mellon University [4]

## Abstract

Hallucinations present a significant challenge in the development and evaluation of large language models (LLMs), directly affecting their reliability and accuracy. While notable advancements have been made in research on textual and visual hallucinations, there is still a lack of a comprehensive benchmark for evaluating auditory hallucinations in large audio language models (LALMs). To fill this gap, we introduce **AHa-Bench**, a systematic and comprehensive benchmark for audio hallucinations. Audio data, in particular, uniquely combines the multi-attribute complexity of visual data with the semantic richness of textual data, leading to auditory hallucinations that share characteristics with both visual and textual hallucinations. Based on the source of these hallucinations, AHa-Bench categorizes them into semantic hallucinations, acoustic hallucinations, and semantic-acoustic confusion hallucinations. In addition, we systematically evaluate seven open-source local perception language models (LALMs), demonstrating the challenges these models face in audio understanding, especially when it comes to jointly understanding semantic and acoustic information. Through the development of a comprehensive evaluation framework, AHa-Bench aims to enhance robustness of LALMs, fostering more reliable and nuanced audio understanding in LALMs.

## 1 Introduction

Large audio-language models (LALM) [8, 7, 10, 51] have demonstrated significant advancements in various tasks, including speech recognition [16], audio classification [28], multimodal understanding [44]. Trained with vast amounts of audio [17] and text data [14], these models [43, 29] have the potential to redefine human-computer interaction [6, 37], facilitating more context-aware and sophisticated systems. However, as these models expand in size and complexity, concerns about their reliability and accuracy have become increasingly prominent [48], particularly in their handling of hallucinations, in cases where the model generates content that is not present in the input data.

Although hallucinations [26] have been extensively investigated in the visual [36, 21] and textual domains [45, 13, 35], where model outputs may diverge from reality or established knowledge, similar research in the audio domain remains relatively underexplored. Given the growing deployment of LALMs in virtual assistants [50, 42], and accessibility tools [27], the robustness of these models in audio understanding has not yet been comprehensively validated. This research gap is particularly concerning, as auditory hallucinations—instances where the model misinterprets, fabricates, or distorts audio inputs—pose significant risks to the integrity of audio-based applications.

---

[*]Equal Contribution.
[†]Corresponding Contribution.

39th Conference on Neural Information Processing Systems (NeurIPS 2025) Track on Datasets and Benchmarks.

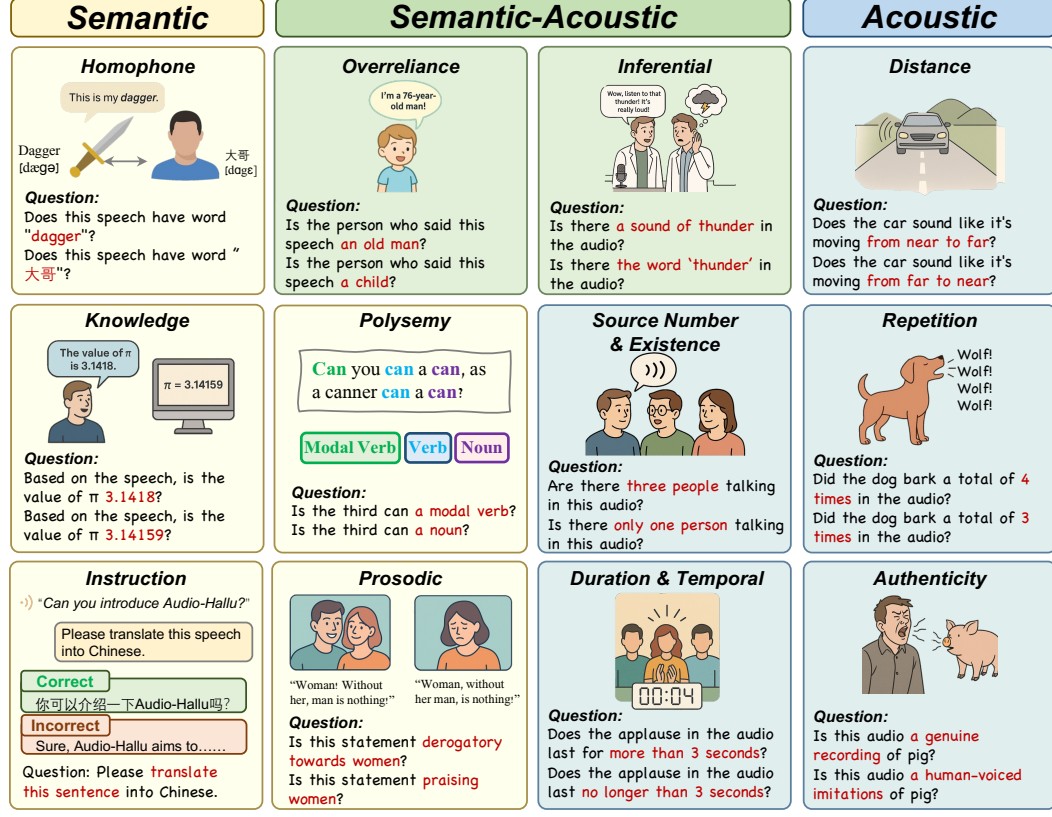

Figure 1: Illustrations of different hallucinations in **AHa-Bench**. Hallucinations can be categorized based on their underlying sources: *Semantic* hallucinations arise from over-reliance on or misinterpretation of semantic information; *Acoustic* hallucinations result from misinterpretation of auditory attributes; and *Semantic-acoustic* hallucinations occur due to confusion between semantic and acoustic information. The red-highlighted text indicates keywords associated with each hallucination type.

To address this gap, we propose AHa-Bench, a comprehensive benchmark specifically designed to evaluate auditory hallucinations in Large Audio-Language Models (LALMs). As illustrated in Figure 1, AHa-Bench systematically categorizes audio hallucinations into three distinct types based on their underlying sources: **(1) Semantic Hallucinations**: Arise when models misinterpret semantic content due to speech-specific attributes such as homophones, prosody, or polysemy, leading to incorrect or ambiguous interpretations. **(2) Acoustic Hallucinations**: Occur when models misinterpret acoustic attributes, such as perceived distance, timbre, or other acoustic characteristics, resulting in incorrect auditory perceptions. **(3) Semantic-Acoustic Hallucinations**: Manifest when models fail to jointly interpret both semantic and acoustic information, causing misalignment or confusion between the two, such as when semantic content is misinterpreted due to conflicting acoustic information. To validate LALMs in audio hallucination challenges, AHa-Bench comprises 396 audio samples and 906 high-quality human-annotated QA pairs, each designed to target a specific hallucination type. We evaluated seven open-source LALMs, assessing them from the perspectives of accuracy, response distribution, and consistency. Our findings reveal substantial challenges in the joint understanding of semantic and acoustic information in existing models.

- We define 14 audio hallucination types, encompassing semantic, acoustic, and semantic-acoustic confusion, establishing a comprehensive taxonomy for diverse auditory scenarios.

- We introduce **AHa-Bench**, a benchmark comprising 396 audio samples and 906 human-annotated QA pairs, systematically designed to assess LALMs' robustness against these hallucinations.

- We evaluate seven open-source LALMs, identifying distinct challenges in jointly interpreting semantic and acoustic information across different model types.

- AHa-Bench exposes critical limitations in existing LALMs, emphasizing the need for more advanced audio understanding capabilities and setting a foundation for future research on mitigating auditory hallucinations.

## 2 Related Works

### 2.1 Large Audio-Language Models

With the rapid development of large language models (LLMs), increasingly powerful large audio-language models [29, 18, 8, 34, 15**?** ] (LALMs) have emerged, demonstrating remarkable capabilities in audio understanding by leveraging massive multimodal corpora. SpeechGPT [53] integrates discrete speech units into LLMs, becoming the first model explicitly centered on speech. Qwen-Audio [8, 7] introduces a comprehensive large-scale audio-language model that covers over 30 tasks, including automatic speech recognition (ASR), speech translation, and audio event detection. To overcome the task-specific overfitting of earlier systems, Salmonn [47] introduces complex story generation tasks, pushing models towards more generalized audio reasoning. Building upon foundational audio understanding, a series of spoken dialogue systems have emerged to support more intelligent and natural human-computer interactions. Futher, some works [39, 49, 32] enhance audio reasoning through the distillation of chain-of-thought (CoT) data.

Despite recent advancements, a critical issue remains largely overlooked: the presence of hallucinations in audio-language models (LALMs). A common failure mode is that LALMs may incorrectly perceive or describe sounds that do not actually exist in the input, posing a significant challenge for deploying these models in real-world scenarios.

### 2.2 Hallucinations in Large Language Models

Hallucinations [25, 2, 22] in large models refer to the generation of fabricated yet seemingly plausible content that the model incorrectly assumes to be true. In the textual modality, hallucinations [25, 33] are typically categorized into two main types: *factual hallucinations* [3], where the generated content contradicts objective facts, and *faithfulness hallucinations*, where the model fails to follow user instructions or maintain consistency with the given context. Building on this foundation, subsequent research [36, 23, 1] has extended hallucination studies to the visual domain, examining whether visual-language models (VLMs) exhibit similar problems. Researchers [36, 9] have identified inconsistencies between model-generated descriptions and actual object properties, leading to the categorization of visual hallucinations into object hallucinations, attribute hallucinations, and relational hallucinations. Further studies [21] have also introduced the notion of "illusions" as a unique subclass of visual hallucinations.

Unlike text and vision, hallucinations in the audio modality exhibit fundamentally distinct characteristics. Audio data combines the multi-attribute complexity of visual data with the semantic richness of text, resulting in auditory hallucinations that share elements with both visual and textual hallucinations. This dual nature introduces novel and more diverse forms of hallucinations that cannot be fully captured by existing taxonomies developed for other modalities.

### 2.3 Audio Hallucination

Several preliminary studies have recently begun to explore the phenomenon of audio hallucinations. For example, Nishimura et al. [40] investigates whether hallucinations can be detected through classification using pretrained audio models. Kuan et al. [30] presents the first study focused on object hallucinations in large audio-language models (LALMs). COMP-A [19] and Match [31] further examine attribute and temporal hallucinations involving overlapping audio events. Meanwhile, AVH-Bench [46] explores the integration of audio signals into multimodal understanding systems as a strategy to mitigate hallucinations in the visual domain. However, current studies have yet to address the unique and nuanced hallucination patterns inherent to the audio modality—such as those arising from semantic ambiguity in speech (e.g., homophones, prosody), misperception of acoustic attributes (e.g., distance, authenticity), or confusion between semantic and acoustic cues (e.g., over-reliance, inferential hallucinations).

To bridge this gap, we propose AHa-Bench, the first comprehensive benchmark for evaluating hallucinations in the audio modality, encompassing 14 distinct types of auditory hallucinations, 396 audio instances, and 906 manually annotated QA pairs. Our benchmark aims to enhance the robustness of large audio-language models (LALMs) by systematically identifying and categorizing hallucinations across both semantic and acoustic dimensions in real-world scenarios.

# 3 Audio Hallucinations in Large Audio-Language Models (LALMs)

Audio information can be broadly categorized into two distinct dimensions: semantic information, referring to specific speech content, and acoustic information, encompassing attributes such as timbre, audio events, and frequency. As illustrated in Figure 1, this work systematically classifies audio hallucinations into three overarching categories:

**I. Semantic Hallucinations.** Semantic hallucinations arise when the model misinterprets the semantic content of speech. These are further subdivided as follows: (1) *Homophone Hallucination*: The model confuses words with similar pronunciations but different meanings (e.g., "hear" vs. "here"). (2) *Polysemy Hallucination*: The model misinterprets words with multiple meanings, selecting an incorrect interpretation based on context. (3) *Prosody Hallucination*: The model misinterprets prosodic cues, leading to errors in sentence segmentation or emphasis. (4) *Instruction Hallucination*: The model erroneously interprets speech as an instruction or query that was not intended by the speaker. (5) *Knowledge Hallucination*: The model generates responses based on outdated, unrelated, or irrelevant knowledge instead of accurately reflecting the current audio context.

**II. Acoustic Hallucinations.** Acoustic hallucinations occur when the model fails to accurately interpret acoustic features, leading to erroneous auditory perceptions. These are categorized as follows: (6) *Existence Hallucination*: The model inaccurately identifies the presence or absence of a specific sound event. (7) *Source Number Hallucination*: The model misjudges the number of sound sources, often due to incorrect acoustic information interpretation. (8) *Distance Hallucination*: The model misinterprets changes in distance based on sound intensity, reverberation, or attenuation. (9) *Duration Hallucination*: The model inaccurately estimates the length of a sound, leading to misinterpretations of its duration. (10) *Temporal Hallucination*: The model confuses the sequence of sound events, causing disordered event perception. (11) *Repetition Hallucination*: The model incorrectly estimates the frequency or repetition of a sound event. (12) *Authenticity Hallucination*: The model fails to distinguish between natural and synthetic sounds, such as genuine human speech versus synthetic imitations.

**III. Semantic-Acoustic Hallucinations.** Semantic-acoustic hallucinations occur when the model over-relies on either semantic or acoustic cues, resulting in conflicting or inferred information. The subcategories are defined as follows: (13) *Overreliance Hallucination*: The model overemphasizes semantic cues, disregarding contradictory acoustic evidence, resulting in misaligned interpretations. (14) *Inferential Hallucination*: The model falsely associates a sound or word not present in the audio, inferred based on related speech or sounds.

# 4 AHA-BENCH: Audio Hallucination Benchmark for LALMs

## 4.1 Evaluation Data Collection

**Stage 1: Audio Collection.** AHa-Bench comprises three distinct audio types: Speech, Sound, and Music. To evaluate semantic-related hallucinations, expert annotators write text content corresponding to each speech sample. Following established practices in prior studies [14, 6], we utilize a TTS model [11] to synthesize highly natural and realistic speech samples for benchmarking. For acoustic-related hallucinations, annotators manually select hallucination-inducing instances from the test sets of existing datasets [17, 19, 4], ensuring that these samples are not part of the training data of the evaluated LALMs.

**Stage 2: Data Annotation.** To facilitate evaluation, we adopt a binary question-answering format as prior work [21]. Let $\mathcal{A} = \{A_1, A_2, \ldots, A_N\}$ denote the set of audio samples. For each audio sample $A_i \in \mathcal{A}$, we construct a corresponding set of $j$ binary questions $\mathcal{Q}_i = \{q_{i,1}, q_{i,2}, \ldots, q_{i,j}\}$. Each audio-question pair $(A_i, q_{i,j})$ is annotated with a binary label $y(A_i, q_{i,j}) \in \{\text{"yes"}, \text{"no"}\}$. To ensure robustness and fairness in evaluation, we maintain a balanced distribution of "yes" and "no" labels across most dataset subsets, thereby mitigating potential biases and minimizing the likelihood of models achieving high accuracy through random guessing or reliance on label priors. For categories with multiple potential pairings, such as polysemy, where a single word may have several possible meanings, we further analyze on each individual pairing, providing a more granular assessment.

**Stage 3: Expert Verification.** To ensure annotation quality, all samples undergo manual verification. For synthesized speech samples, we primarily assess whether the audio content aligns with the

| Hallucination | #Inst. | #QAPs. | #Yes | #No |
|---|---|---|---|---|
| *Semantic Hallucination* | | | | |
| Homophone | 35 | 70 | 35 | 35 |
| Polysemy | 17 | 102 | 17 | 85 |
| Prosodic | 28 | 56 | 28 | 28 |
| Knowledge | 32 | 64 | 32 | 32 |
| Instruction | 20 | 20 | - | - |
| *Acoustic Hallucination* | | | | |
| Source Number | 20 | 40 | 20 | 20 |
| Existence | 42 | 84 | 42 | 42 |
| Distance | 16 | 48 | 16 | 32 |
| Duration | 20 | 40 | 20 | 20 |
| Temporal | 26 | 104 | 52 | 52 |
| Repetition | 20 | 40 | 20 | 20 |
| Authenticity | 60 | 120 | 60 | 60 |
| *Semantic-Acoustic Confusion Hallucination* | | | | |
| Inferential | 44 | 88 | 44 | 44 |
| Overreliance | 16 | 32 | 16 | 16 |
| AHa-Bench (Total) | 396 | 906 | 402 | 484 |

(a) Statistics of each audio hallucinations.

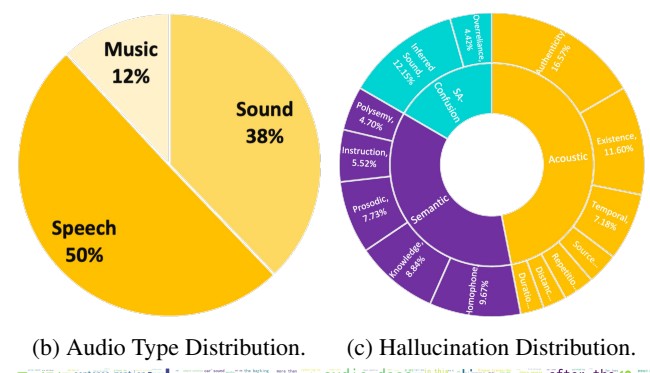

(b) Audio Type Distribution.     (c) Hallucination Distribution.

(d) Word Cloud of AHa-Bench.

Figure 2: Detailed Statistical Analysis of AHa-Bench. **#Inst.**: The number of audio instances. **#QAPs.**: Total number of QA pairs. **#Yes/ #No**: Number of questions answered as Yes/ No.

Table 1: Comparison of Multi-Modal Hallucination Benchmarks. **#QAPs.**: Total number of QA pairs. **#H-Edited**: Number of manually verified QA pairs. **#Inst.**: Number of image or audio instances. The numbers in brackets represent the number of hallucination types evaluated.

| Benchmarks | #QAPs. | #H-Edited | #Inst. | Hallucination Types | | |
|---|---|---|---|---|---|---|
| | | | | **Acoustic** | **Semantic** | **SA-Confusion** |
| *Visual Hallucination Benchmarks* | | | | | | |
| POPE [36] | 3,000 | 0 | 500 | - | - | - |
| GAVIE [38] | 1,000 | 0 | 1,000 | - | - | - |
| Bingo [9] | 370 | 370 | 308 | - | - | - |
| HallusionBench [21] | 1,129 | 1,129 | 346 | - | - | - |
| *Audio Hallucination Benchmarks* | | | | | | |
| Audio Hallucination [30] | 30K | 0 | 1,000 | ✓(1) | ✗ | ✗ |
| CompA-Order [19] | 900 | 0 | 400 | ✓(1) | ✗ | ✗ |
| MATCH [31] | 15.5K | 0 | 960 | ✓(3) | ✗ | ✗ |
| AVHBENCH [46] | 5.3K | 0 | 1,124 | ✓(1) | ✗ | ✗ |
| AHa-Bench (ours) | 906 | 906 | 396 | ✓(7) | ✓(5) | ✓(2) |

intended attributes in terms of timbre, prosody, pronunciation, and semantic content. For samples sourced from AudioSet, annotators incorporate visual context during verification to accurately label attributes such as source distance, repetition frequency, and source count. Similarly, for samples drawn from other open datasets, we implement a rigorous verification process. During the review process, researchers also verify the alignment between the audio content and the associated QA pairs, ensuring consistency and accuracy across all samples.

## 4.2 Dataset Statistics

**Detailed Statistical Analysis of AHa-Bench.** As shown in Table 2a, we present a comprehensive statistical analysis of AHa-Bench, detailing the distribution of various hallucination types. To ensure data balance, we systematically collected a sufficient number of audio samples for each hallucination category. Each sample is associated with multiple corresponding questions to facilitate comprehensive evaluation. To further assess the representation of different audio types within AHa-Bench, the dataset includes three distinct audio categories: Music, Sound, and Speech, as illustrated in Figure 2b. The pie chart in Figure 2c visualizes the distribution of samples across these hallucination types, demonstrating a relatively balanced representation across categories. Figure 2d presents a word cloud that highlights key terms and concepts in AHa-Bench.

**Comparison with Other Hallucination Datasets.** Previous audio hallucination datasets [31, 19] primarily focused on evaluating object and temporal hallucinations, relying on acoustic information to determine the presence of specific sounds or the temporal relationships between multiple audio events. In contrast, AHa-Bench adopts a more comprehensive approach, examining a broader range of hallucination types across multiple acoustic dimensions. Furthermore, AHa-Bench is the first benchmark to explicitly emphasize both semantic hallucinations and semantic-acoustic confusion hallucinations, areas that have been largely overlooked in previous benchmarks. This extensive coverage enables a more nuanced assessment of model performance, facilitating a deeper understanding of how LALMs handle diverse audio hallucinations.

In addition, unlike previous datasets that primarily relied on predefined templates and audio labels to construct samples in a batch processing manner, AHa-Bench employs a more rigorous annotation process. Every sample and question pair is meticulously crafted by human experts, ensuring precise evaluation for each hallucination type. In terms of dataset scale, AHa-Bench aligns with other high-quality visual hallucination benchmarks [9, 21], providing a sufficiently large sample size to support reliable and robust experimental conclusions.

# 5 Benchmarking Audio Hallucinations in Large Audio-Language Models

## 5.1 Compared Large Audio Language Models

We conduct extensive experiments on our Aha-Bench to evaluate a total of 7 LALMs, including SALMONN-13B [47], Qwen-Audio [8], Qwen2-Audio [7], Qwen2-Audio-Instruct [7], GLM4-Voice [52], Kimi-Audio [10] and Gemini-2.5-Pro (Preview 05-06). Additionally, we include Random Chance (i.e., randomly choosing 'Yes' or 'No') as a baseline. The model detailed description and evaluation prompt template can be found in Appendix A and Appendix C.1.

## 5.2 Evaluation Suite

**GPT4-Assisted Evaluation.** Due to the high diversity in the responses generated by Large Audio-Language Models (LALMs), we refer to prior work [12] and use GPT-4o [41] to preprocess the answers, categorizing them into three possible responses: '*Yes*', '*No*', and '*Unknown*'. The introduction of the '*Unknown*' option ensures that GPT-4o can handle uncertainty and provides insight into the frequency with which the model opts for this neutral response, rather than forcing a '*Yes*' or '*No*' answer when it is unsure. This approach helps avoid potential biases in model behavior when faced with ambiguous or uncertain inputs. Detailed prompt templates for this process can be found in Appendix C.2. Additionally, for Instruction Hallucination, we first calculate the Word Error Rate (WER) for each generated response. If the WER is less than 10%, the response is classified as '*Yes*'; otherwise, it is classified as '*No*'.

**Correctness Assessment.** The accuracy (ACC) metric is used to assess the correctness of LALMs responses to binary audio-question pairs. To mitigate the possibility of random guessing by LALMs, we adopt a stricter evaluation criterion. Following prior work [21], we define a response as correct only if all question pairs associated with an audio instance are answered consistently and correctly. The accuracy metric is calculated as follows:

$$\text{ACC}_i = \frac{\sum_{j=1}^{|\mathcal{A}_i|} \mathbb{1}\left(\forall q \in \mathcal{Q}_{i,j}, \ \hat{y}(A_{i,j}, q) = y(A_{i,j}, q)\right)}{|\mathcal{A}_i|}, \tag{1}$$

where $\mathcal{A}_i$ denotes the set of audio instances for the $i$-th hallucination type, $\mathcal{Q}_{i,j}$ represents the set of questions associated with the $j$-th audio instance in $\mathcal{A}_i$, $\hat{y}(A_{i,j}, q)$ is the model's predicted response for question $q$ and $y(A_{i,j}, q)$ is the ground truth response for question $q$.

**Yes/No Bias Test.** According to previous hallucination researches [36, 21], some models [20] exhibit a tendency to respond with "Yes" in most cases. If a model demonstrates a strong bias or inclination to provide a particular response regardless of the actual question, further analysis may not be necessary. We introduce the Yes/No Bias Score ($\text{Bias}_{\text{Y/N}}$) to evaluate the model's tendency to favor "Yes" or "No" responses. Following the prior work [24], we define the bias as the difference between the False Positive Rate (FPR) and False Negative Rate (FNR):

$$\text{Bias}_{\text{Y/N}} = \frac{\mathbb{1}\left[\hat{y}(A_i, q_{i,j}) = \text{Yes}\right]}{\mathbb{1}\left[y(A_i, q_{i,j}) = \text{No}\right]} - \frac{\mathbb{1}\left[\hat{y}(A_i, q_{i,j}) = \text{No}\right]}{\mathbb{1}\left[y(A_i, q_{i,j}) = \text{Yes}\right]}, \tag{2}$$

Table 2: Comparison of Accuracy on AHa-Bench. *Homo.*: Homophone, *Poly.*: Polysemy, *Proso.*: Prosodic, *Knowl.*: Knowledge, *Instr.*: Instruction, *SrcNum.*: Source Number, *Exist.*: Existence, *Dist.*: Distance, *Dur.*: Duration, *Temp.*: Temporal, *Repet.*: Repetition, *Auth.*: Authenticity, *Inf$_A$.*: Inferential from acoustic information, *Inf$_S$.*: Inferential from semantic information, *Overrel.*: Overreliance. **Best-performing model** is marked in bold, and second-best model is underlined.

| Models | Semantic Hallucination | | | | | Acoustic Hallucination | | | | | | | SA-Confusion | | | Mean |
|---|---|---|---|---|---|---|---|---|---|---|---|---|---|---|---|---|
| | Homo. | Poly. | Proso. | Knowl. | Instr. | SrcNum. | Exist. | Dist. | Dur. | Temp. | Repet. | Auth. | Inf$_a$. | Inf$_s$. | Overrel. | |
| Random | 24.64 | 5.15 | 23.08 | 23.05 | - | 23.96 | 25.00 | 9.72 | 30.56 | 12.50 | 23.96 | 24.37 | 16.96 | 22.50 | 25.00 | 19.36 |
| *Open-Source LALMs* | | | | | | | | | | | | | | | | |
| GLM4-Voice | 56.79 | 0.00 | 30.29 | 50.00 | 0.00 | 2.63 | 7.14 | **12.50** | 25.62 | 3.85 | 20.00 | 1.87 | 0.00 | 20.83 | 14.84 | 16.42 |
| SALMONN | 12.50 | 0.00 | 25.00 | 37.89 | - | 0.00 | 18.75 | 0.00 | 0.00 | 0.00 | 0.00 | 0.00 | 3.57 | 0.00 | 18.75 | 7.76 |
| Qwen-Audio | 39.64 | 0.74 | 26.92 | 27.73 | 30.00 | 17.76 | 28.87 | 9.38 | 25.00 | 13.46 | 21.88 | 19.79 | 13.39 | **27.92** | 34.38 | 22.46 |
| Qwen2-Audio | 26.79 | 0.00 | 20.19 | 40.62 | 45.00 | 0.00 | **51.49** | 0.00 | 0.00 | 0.00 | 3.12 | 14.58 | 9.82 | 1.67 | 28.91 | 16.15 |
| Qwen2-Audio-Inst | 22.14 | 0.00 | 20.19 | 59.38 | 75.00 | 0.00 | 33.63 | 0.00 | 10.00 | 0.00 | 15.00 | 23.75 | 8.04 | 0.00 | 43.75 | 20.73 |
| FunAudioLLM | 56.43 | 15.44 | 16.35 | 48.44 | 100.00 | 22.37 | 5.65 | 0.00 | 40.62 | 8.17 | 14.37 | 21.25 | 0.00 | 10.00 | 21.88 | 20.54 |
| Kimi-Audio | 47.14 | 5.88 | 22.60 | 44.53 | **90.00** | 5.26 | 46.73 | 0.00 | 16.87 | 2.88 | 8.75 | 5.83 | 5.36 | 3.33 | **53.91** | 23.94 |
| *Closed-Source LALMs* | | | | | | | | | | | | | | | | |
| GPT-Audio | 42.14 | 4.41 | 17.31 | 49.22 | 28.75 | 39.47 | 3.75 | 0.00 | 13.75 | 1.92 | 23.75 | 28.75 | 64.29 | 10.00 | 7.81 | 25.36 |
| Gemini-2.5-Pro | **62.86** | **23.53** | **42.31** | **81.25** | 60.00 | **36.84** | 47.62 | 6.25 | **60.00** | **30.77** | 40.00 | **41.67** | **78.57** | 13.33 | 50.00 | **45.00** |

where Bias$_{Y/N}$ ≈ 0 indicates balanced responses, while values approaching -1 or 1 indicate strong biases toward "No" or "Yes," respectively.

**Consistency Test.** To assess the consistency of model responses, we introduce the Diff metric, which quantifies the proportion of audio instances for which the model's responses exhibit logical inconsistency. This metric is defined as the proportion of audio instances where the model's responses to the associated set of questions are neither entirely correct nor entirely incorrect. Following prior work, the Diff is defined as:

$$\text{Diff}_i = \frac{\sum_{j=1}^{|\mathcal{A}_i|} \mathbb{1}\left(0 < \sum_{q \in \mathcal{Q}_{i,j}} \mathbb{1}\left(\hat{y}(A_{i,j}, q) = y(A_{i,j}, q)\right) < |\mathcal{Q}_{i,j}|\right)}{|\mathcal{A}_i|}. \tag{3}$$

**Robustness Guarantee.** Due to the inherent stochasticity of large language models (LLMs), the variability in output across different sampling runs can potentially impact the robustness of evaluation systems [12]. To mitigate the effect of sampling variability on performance assessment, we performed 16 sampling runs for each question to enhance the stability and reliability of the evaluation results.

## 5.3 Main Results

We present the hallucination performance of different models on AHa-Bench in Table 2, Table 3 and Table 4, with additional experimental results provided in Appendix D. From the analysis of the experimental results, we derive the following key observations:

### 5.3.1 Audio Hallucination Challenges Across Different LALMs

LALMs can be broadly categorized into audio understanding models, dialogue models, and hybrid models, each exhibiting distinct hallucination patterns:

- **Audio Understanding Models** (e.g., Qwen-Audio, Qwen2-Audio): These models excel in acoustic hallucination tasks, with Qwen2-Audio achieving 51.49% accuracy in Existence Hallucination and Qwen-Audio performing well in Source Number (17.76%), Inferential from semantic information (27.92%), and Duration (25.00%). However, their focus on acoustic attributes limits their performance in prosodic tasks, where semantic comprehension is essential.
- **Dialogue Models** (e.g., GLM4-Voice): Primarily trained on speech semantics, these models perform well in semantic hallucination tasks but are prone to instruction hallucinations, often misinterpreting general speech as commands.
- **Hybrid Models** (e.g., Kimi-Audio, Qwen2-Audio-Instruct): Trained on both dialogue and audio tasks, these models face challenges in semantic-acoustic confusion hallucinations, where semantic cues and acoustic events conflict. Despite this, Kimi-Audio demonstrates effective mitigation of Overreliance Hallucinations, achieving 53.91% accuracy by distinguishing commands from general speech, indicating that targeted training can reduce such hallucinations.

Table 3: Yes/No Bias Analysis. The Bias$_{Y/N}$ metric ($\sim 0$) assesses response bias toward "yes" or "no" answers. Results with minimal bias are highlighted in green, while results with the greatest bias are marked in red. The mean score is the average of the absolute Bias$_{Y/N}$ across different hallucinations.

| Models | Semantic Hallucination | | | | Acoustic Hallucination | | | | | | | SA-Confusion | | | Mean |X| |
|---|---|---|---|---|---|---|---|---|---|---|---|---|---|---|---|
| | Homo. | Poly. | Proso. | Knowl. | SrcNum. | Exist. | Dist. | Dur. | Temp. | Repet. | Auth. | Inf$_a$. | Inf$_s$. | Overrel. | |
| *Open-Source LALMs* | | | | | | | | | | | | | | | |
| GLM4-Voice | -0.09 | 0.13 | -0.01 | 0.03 | 0.67 | 0.74 | 0.16 | 0.01 | 0.02 | 0.20 | 0.80 | 1.00 | 0.42 | -0.64 | 0.25 |
| SALMONN | 0.88 | 0.92 | 0.62 | 0.22 | 1.00 | 0.84 | 0.50 | 1.00 | 0.95 | 1.00 | 1.00 | 0.96 | 1.00 | 0.63 | 0.82 |
| Qwen-Audio | -0.21 | -0.64 | -0.42 | -0.45 | -0.04 | -0.07 | -0.16 | -0.06 | 0.04 | -0.04 | -0.11 | 0.15 | 0.14 | -0.34 | -0.16 |
| Qwen2-Audio | 0.56 | 0.80 | 0.58 | 0.42 | 0.80 | 0.33 | 0.62 | 0.63 | 0.74 | 0.87 | 0.19 | 0.69 | 0.98 | 0.08 | 0.59 |
| Qwen2-Audio-Inst | 0.78 | 0.75 | 0.39 | 0.01 | 1.00 | 0.67 | 0.45 | 0.86 | 0.82 | 0.55 | 0.70 | 0.74 | 0.80 | 0.00 | 0.61 |
| FunAudioLLM | | | | | | | | | | | | | | | 0.11 |
| Kimi-Audio | 0.40 | 0.66 | 0.27 | 0.12 | 0.63 | 0.48 | 0.63 | -0.20 | 0.77 | 0.81 | 0.84 | 0.95 | 0.91 | 0.25 | 0.54 |
| *Closed-Source LALMs* | | | | | | | | | | | | | | | |
| Gemini-2.5-Pro | 0.34 | 0.09 | 0.15 | -0.12 | 0.20 | 0.52 | 0.38 | 0.10 | 0.29 | -0.05 | 0.18 | 0.07 | 0.60 | 0.07 | 0.20 |

Table 4: Consistency Analysis. The Diff metric quantifies the proportion of audio instances where the model's responses exhibit logical inconsistency. A lower value indicates better consistency in model outputs. The **best-performing model** is marked in bold, and the second-best model is underlined.

| Models | Semantic Hallucination | | | | Acoustic Hallucination | | | | | | | SA-Confusion | | | Mean |
|---|---|---|---|---|---|---|---|---|---|---|---|---|---|---|---|
| | Homo. | Poly. | Proso. | Knowl. | SrcNum. | Exist. | Dist. | Dur. | Temp. | Repet. | Auth. | Inf$_a$. | Inf$_s$. | Overrel. | |
| *Open-Source LALMs* | | | | | | | | | | | | | | | |
| GLM4-Voice | 40.71 | 100.00 | **44.71** | **18.75** | 86.19 | 86.31 | 81.25 | 56.88 | 94.23 | **40.00** | 92.71 | 100.00 | 66.67 | 69.54 | 65.20 |
| SALMONN | 87.5 | 100.00 | 68.75 | 52.73 | 100.00 | 81.25 | 87.50 | 100.00 | 100.00 | 100.00 | 100.00 | 96.43 | 100.00 | 68.75 | 82.86 |
| Qwen-Audio | 45.36 | 99.26 | 50.96 | 59.77 | **59.87** | 46.43 | 67.18 | 46.25 | 84.14 | 73.12 | 61.46 | 65.18 | **50.00** | 27.34 | 55.75 |
| Qwen2-Audio | 65.35 | 100.00 | 70.19 | 53.13 | 100.00 | 48.51 | 98.44 | 77.50 | 96.15 | 96.88 | 71.25 | 90.18 | 98.33 | 48.43 | 74.29 |
| Qwen2-Audio-Inst | 77.86 | 100.00 | 62.50 | 28.51 | 100.00 | 66.37 | 88.28 | 85.63 | 98.08 | 85.00 | 69.79 | 88.39 | 79.58 | 18.75 | 69.91 |
| Kimi-Audio | 50.36 | 94.12 | 59.61 | 29.30 | 94.08 | 53.27 | 100.00 | 51.26 | 97.12 | 91.25 | 93.96 | 94.64 | 91.25 | 28.90 | 68.61 |
| *Closed-Source LALMs* | | | | | | | | | | | | | | | |
| Gemini-2.5-Pro | **34.28** | **76.47** | 46.15 | 18.75 | 63.16 | 52.38 | 87.50 | **30.00** | 69.23 | 55.00 | 55.00 | 21.43 | 66.67 | **12.50** | 45.90 |

## 5.4 Yes/No Bias and Response Consistency Analysis

Table 3 presents a comprehensive evaluation of yes/no bias and response consistency across the evaluated models. Instruction-following models, such as Qwen2-Audio-Instruct (0.61), exhibit a noticeable affirmative bias, increasing the likelihood of false positives. The most pronounced affirmative bias is observed in SALMONN (0.82), indicating a strong tendency to over-confirm responses. This over-confirmation tendency also contributes to inconsistent responses, as highlighted in Table 4. Regarding response consistency, Qwen-Audio achieves the lowest inconsistency score (55.75%), maintaining stable but somewhat rigid response patterns. In contrast, dialogue-oriented models like Qwen2-Audio-Instruct and SALMONN exhibit higher variability, particularly in SA-Confusion tasks, suggesting a trade-off between conversational flexibility and response stability.

This inconsistency partially stems from semantic hallucinations, as models occasionally default to affirmative responses based on the question content rather than actual audio cues. Notably, Gemini demonstrates the highest consistency, indicating its stronger resistance to semantic hallucinations. We further explore language hallucinations in LALMs when understand audio content in Appendix D.3.

## 5.5 Mitigating Audio Hallucinations in LALMs

Table 2 highlights the significant challenge of audio hallucinations, with some models performing even worse than random guessing. Despite this, Qwen-Audio and Kimi-Audio demonstrate relatively robust performance. The closed-source Gemini-2.5-pro even achieved 45% accuracy.

Qwen-Audio excels at handling authenticity and reasoning tasks, which can be attributed to its task-centered training approach, effectively reducing unnecessary interference. Kimi-Audio shows the strongest resistance to hallucinations, ranking in the top two among open-source LALMs in 6 out of 14 hallucinations, with notably high accuracy in instruction hallucination (90.00%) and over-reliance hallucination (53.91%). The results reveal a trade-off between response consistency and conversational adaptability. Structured models like Qwen-Audio maintain consistent but somewhat rigid response patterns, while dialogue-oriented models like Qwen2-Audio-Instruct and SALMONN

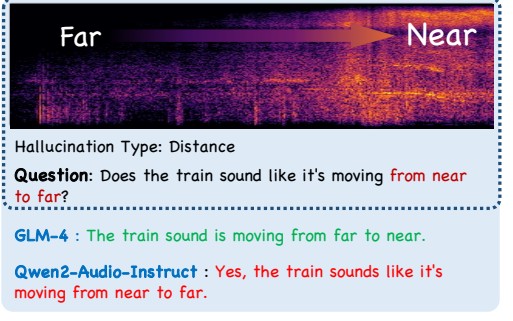

Hallucination Type: Distance

**Question**: Does the train sound like it's moving from near to far?

GLM-4 : The train sound is moving from far to near.

Qwen2-Audio-Instruct : Yes, the train sounds like it's moving from near to far.

(a) Sample 1: Distance Hallucination Example.

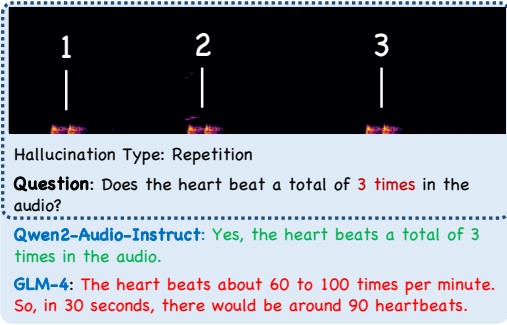

Hallucination Type: Repetition

**Question**: Does the heart beat a total of 3 times in the audio?

Qwen2-Audio-Instruct: Yes, the heart beats a total of 3 times in the audio.

GLM-4: The heart beats about 60 to 100 times per minute. So, in 30 seconds, there would be around 90 heartbeats.

(b) Sample 2: Repetition Hallucination Example.

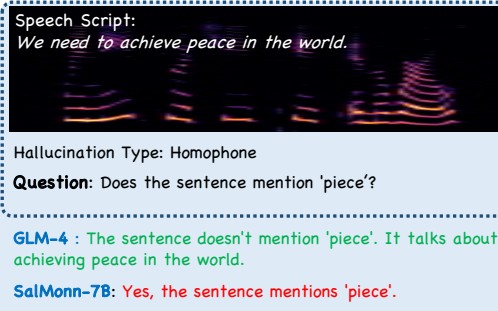

Speech Script:
*We need to achieve peace in the world.*

Hallucination Type: Homophone

**Question**: Does the sentence mention 'piece'?

GLM-4 : The sentence doesn't mention 'piece'. It talks about achieving peace in the world.

SalMonn-7B : Yes, the sentence mentions 'piece'.

(c) Sample 3: Homophone Hallucination Example.

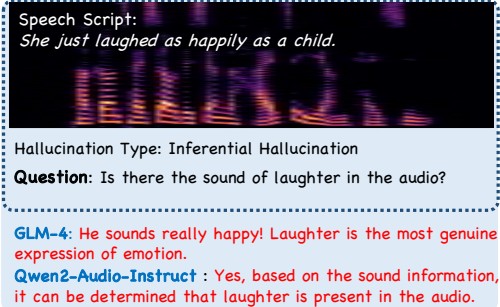

Speech Script:
*She just laughed as happily as a child.*

Hallucination Type: Inferential Hallucination

**Question**: Is there the sound of laughter in the audio?

GLM-4: He sounds really happy! Laughter is the most genuine expression of emotion.

Qwen2-Audio-Instruct : Yes, based on the sound information, it can be determined that laughter is present in the audio.

(d) Sample 4: Inferential Hallucination Example.

Figure 3: Failure cases on AHa-Bench. Correct responses are highlighted in green, while incorrect responses exhibiting hallucinations are marked in red.

offer greater conversational flexibility, but at the cost of increased affirmative bias. This emphasizes the importance of avoiding unnecessary hallucinations during conversations. Gemini-2.5-pro demonstrates the strongest performance against language hallucinations, partly due to its effective instruction comprehension, which avoids affirmative responses. This suggests that balancing semantic generalization with strict instruction adherence is also a key strategy for mitigating hallucinations in complex auditory contexts.

### 5.6 Qualitative Comparison

In Table 3, we present several failure cases of various LALM models on AHa-Bench, offering qualitative insights into how these models handle auditory hallucinations. Specifically, Figure 3a and Figure 3b illustrate examples of acoustic hallucinations, Figure 3c showcases a semantic hallucination, and Figure 3d demonstrates a semantic-acoustic hallucination. These examples not only underscore the specific challenges encountered by LALM models but also provide readers with a more nuanced understanding of the dataset structure and the subtle distinctions between hallucination types. Additionally, we present more diverse hallucination cases in Appendix E, further illustrating the range and complexity of auditory hallucinations in LALM models.

## 6 Conclusion

In this study, we introduce AHa-Bench, a comprehensive benchmark specifically designed to systematically assess audio hallucinations in large audio-language models (LALMs). AHa-Bench categorizes these hallucinations into semantic, acoustic, and semantic-acoustic confusion types, encompassing 14 distinct categories. It includes 396 audio samples and 906 human-annotated QA pairs, meticulously crafted to evaluate LALMs' robustness against these hallucinations. Through a systematic evaluation of seven open-source LALMs, we highlight the significant challenges these models encounter in accurately interpreting complex audio content. Moreover, our analysis uncovers differential vulnerabilities across LALM architectures, demonstrating how specific hallucination types disproportionately impact certain model designs. By establishing a structured framework for assessing audio hallucinations, AHa-Bench emerges as a critical resource for enhancing the robustness and reliability of LALMs, promoting more nuanced and accurate audio understanding.

## Acknowledgements

This work was supported in part by National Key R&D Program of China (No. 2022ZD0162000) and National Natural Science Foundation of China (No. 62222211, U24A20326, 624B2128).

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

## A  Model Details

**GLM4-Voice [52]**   GLM4-Voice is an end-to-end spoken dialogue model trained on extensive conversational data, enabling real-time speech interaction. It adopts interleaved and parallel decoding strategies to simultaneously generate text and audio tokens, effectively supporting low-latency dialogue systems.

**SALMONN-13B [47]**   SALMONN-13B is a multimodal large language model designed to process and understand speech, audio events, and music, representing a significant advancement in generalized auditory capabilities for LLMs. It demonstrates exceptional performance in speech recognition, audio captioning, and speech translation, while also generalizing to tasks such as slot filling, keyword extraction, and multilingual speech translation. Notably, SALMONN-13B exhibits emergent abilities in audio-based storytelling and speech-audio co-reasoning.

**Qwen-Audio [8]**   Qwen-Audio is a large-scale audio language model that supports diverse audio types, languages, and tasks. It achieves state-of-the-art performance across multiple benchmarks, showcasing universal audio understanding capabilities.

**Qwen2-Audio [7]**   Qwen2-Audio builds upon Qwen-Audio, integrating audio and text inputs to generate textual outputs. It demonstrates state-of-the-art performance in instruction-following capabilities across speech, sound, music, and mixed-audio subsets, highlighting its proficiency in audio understanding and dialogue.

**Qwen2-Audio-Instruction [7]**   Qwen2-Audio-Instruction, based on Qwen2-Audio, is designed to engage in dialogue with users regarding audio and text-based inquiries. Trained extensively on spoken dialogue data, it exhibits enhanced communication and conversational abilities.

**Kimi-Audio [10]**   Kimi-Audio is a comprehensive large audio language model based on Qwen2.5-7B, designed to perform audio understanding, generation, and conversation tasks within a unified architecture. It achieves the highest scores in emotion control, empathy, and speed control, underscoring its proficiency in generating expressive and controllable speech.

Table 5: Sources of Audio Instances for Evaluating Different Types of Hallucinations.

| Hallucination Type | Hallucination Name | Audio Instance Source |
|---|---|---|
| Semantic Hallucination | Homophone Hallucination | TTS Generated |
| | Polysemy Hallucination | TTS Generated |
| | Prosodic Hallucination | TTS Generated |
| | Kknowledge Hallucination | TTS Generated |
| | Instruction Hallucination | TTS Generated |
| Acoustic Hallucination | Existence Hallucination | AudioSet Test Set |
| | Source Number Hallucination | AudioSet Test Set |
| | Duration Hallucination | AudioSet Test Set |
| | Temporal Hallucination | Comp-A |
| | Distance Hallucination | AudioSet Test Set |
| | Repetition Hallucination | AudioSet Test Set |
| | Authenticity Hallucination | Vocalsketch |
| Semantic-Acoustic Hallucination | Inferential Hallucination (Speech) | TTS Generated |
| | Inferential Hallucination (Sound) | AudioSet Test Set |
| | Over-Reliance Hallucination | TTS Generated |

**Gemini 2.5 Pro**  [3] Gemini 2.5 Pro is the latest generation of large language models launched by Google DeepMind. It is designed to handle complex reasoning tasks and has strong multi-modal understanding and programming capabilities.

# B  DATASET DETAILS

## B.1  Annotator Details

A total of four experts participated in the Expert Review stage. Each domain (semantic hallucination, acoustic hallucination and semantic-acoustic hallucination) was assigned one expert for both the annotation, filtering and review stages. The group consisted of three males and one female. The experts involved in the Expert Annotation stage were MS/PhD students with a strong foundational understanding of their respective domains. For the Expert Review stage, the annotators included PhD students and industry practitioners, whose expertise was validated through their published research and contributions to the field. These experts brought substantial domain knowledge and research experience to the project. They possess a comprehensive understanding of sound analysis and are adept at identifying subtle audio details. Their expertise is both technical and theoretical, enabling them to approach the annotation process with a nuanced perspective. This background allowed them to handle complex audio data with precision, ensuring that the annotations were both accurate and meaningful. The collective experience of these experts significantly enhanced the quality and reliability of the annotated audio corpus, contributing to a robust and well-curated dataset.

## B.2  Data Source for Different Audio Hallucinations.

Table 5 presents the data sources for each hallucination subset, providing a comprehensive overview to facilitate reader understanding.

## B.3  Source Dataset Details.

**AudioSet [17]**  AudioSet is a large-scale dataset comprising over 2 million audio clips, each annotated with one or more of 527 audio event classes encompassing a broad spectrum of everyday sounds. Developed by Google, it is constructed using 10-second segments from YouTube videos, providing a comprehensive representation of environmental sounds, music, speech, and various audio events. Each audio clip is labeled using a hierarchical ontology, enabling both fine-grained

---
[3]https://deepmind.google/technologies/gemini/pro/

| **Model Evaluation Prompt Template for Qwen2-Audio-Instruct.** |
|---|
| **System**: You are a helpful assistant. 
 **User**: {<Audio Instance>} Listen to the given audio carefully and answer this question: {Question} 

 **Assistant**: |

Table 6: Model Evaluation Prompt Template for Qwen2-Audio-Instruct.

and coarse-grained sound categorization. To ensure that the data in AHa-Bench remains unseen by the evaluated models, we exclusively use the test set of AudioSet as the data source, effectively minimizing potential data leakage and maintaining the integrity of the evaluation process.

**VocalSketch [5]**  VocalSketch contains thousands of vocal imitations of a large set of diverse sounds. These imitations were collected from hundreds of contributors via Amazon's Mechanical Turk website. The dataset also contains data on hundreds of people's ability to correctly label these vocal imitations, also collected via Amazon's Mechanical Turk. This data set will help the research community understand which audio concepts can be effectively communicated with this approach.

**CompA-Order [19]**  CompA-order is constructed using the test set of AudioSet to assess the capability of Large Audio-Language Models (LALMs) to understand the temporal order of multiple acoustic events. Each acoustic event within an audio clip can either succeed, precede, or occur simultaneously with another event. CompA-order consists of 400 test instances, each containing at least two audio-caption pairs. In each pair, the audio clips include the same two acoustic events, but with their order of occurrence intentionally varied, enabling a targeted evaluation of the model's ability to discern temporal sequencing.

## C   Evaluation Details

### C.1   Model Evaluation Prompt Template

We present the Model Evaluation Prompt Template, drawing on the evaluation kit[4] established in Kimi-Audio [10]. Since different models utilize distinct prompt templates, we present the Qwen2-Audio-Instruct template as an example for clarity in Figure 6. For additional prompt templates used for other models, please refer to the GitHub repository and the supplementary materials.

### C.2   GPT-4 Assisted Evaluation Prompt Template

We presented the gpt prompt in Table 7.

## D   More Experimental Results

### D.1   Instance-level Accuracy

In Table 8, we present the comparison of instance-level accuracy on AHa-Bench. Unlike the accuracy reported in the main text, which requires logical consistency across multiple questions, the accuracy here is evaluated at the instance level, where a response is considered correct as long as any single question is answered correctly.

### D.2   Language Hallucination in LALMs

Language hallucinations can significantly impact the capabilities of Multimodal Large Language Models (MLLMs). To gain a more detailed understanding of the extent to which these hallucinations arise from the audio modality, we further explore the language hallucinations present when different

---

[4]`https://github.com/MoonshotAI/Kimi-Audio-Evalkit`

---
**GPT-4 Assisted Evaluation Prompt Template.**

**Task:** You are an AI assistant responsible for assessing the alignment of an answer with three predefined response options: **Yes, No, Unknown**.
Your objective is to evaluate the given question-answer pair and determine which of the three options (Yes, No, Unknown) best represents the answer.
- If the answer clearly aligns with an affirmative response, output **'Yes'**.
- If the answer clearly aligns with a negative response, output **'No'**.
- If the answer is ambiguous or does not sufficiently align with either Yes or No, output **'Unknown'**.

Your output must consist of a single word: **'Yes', 'No', or 'Unknown'**.

**Examples:**

1. **Question:** Is the car moving fast?
**Answer:** Yes, it is speeding down the highway.
**Output: Yes**

2. **Question:** Is the dog barking?
**Answer:** The dog is lying quietly on the floor.
**Output: No**

3. **Question:** Is it raining outside?
**Answer:** I don't hear any rain, but it could have rained earlier.
**Output: Unknown**

**Input:**
**Question:** {question}
**Answer:** {answer}
**Output:**

---

Table 7: GPT-4 Assisted Evaluation Prompt Template.

LALMs interpret audio. As noted by Work [21], when the same question is posed, but the audio instances differ yet the answers remain the same, this indicates the presence of language hallucinations in the LALM.

As shown in Table 9, we conducted an analysis on several hallucination categories in AHa-Bench using paired audio data to assess the proportion of language hallucinations in different models. This investigation provides valuable insights into the frequency and impact of language hallucinations in the audio understanding process across LALMs.

Qwen-Audio and Qwen2-Audio, as audio understanding models, achieved the lowest language hallucination rates, thanks to their relatively rigid response patterns, even outperforming Gemini-2.5-Pro (0.45). Despite Gemini-2.5-Pro having a higher frequency of language hallucinations, it demonstrates better performance in audio hallucination tasks compared to Qwen-Audio, suggesting that solving the challenge of audio hallucinations requires not only resistance to language hallucinations but also a strong ability to counter audio-based hallucinations.

### D.3 Error bar in LALMs

To obtain more robust experimental results, we conducted multiple trials on AHa-Bench and calculated the average values. In Figure 4, we present box plots showing the accuracy (acc) of different models across various audio hallucination categories. This approach provides a clear visualization of the models' performance and variability in handling audio hallucinations.

Table 8: Comparison of Instance-level Accuracy on AHa-Bench. *Homo.*: Homophone, *Poly.*: Polysemy, *Proso.*: Prosodic, *Knowl.*: Knowledge, *Instr.*: Instruction, *SrcNum.*: Source Number, *Exist.*: Existence, *Dist.*: Distance, *Dur.*: Duration, *Temp.*: Temporal, *Repet.*: Repetition, *Auth.*: Authenticity, *Inf*$_A$.: Inferential from acoustic information, *Inf*$_S$.: Inferential from semantic information, *Overrel.*: Overreliance. **Best-performing model** is marked in bold, and second-best model is underlined.

| Models | Semantic Hallucination | | | | | Acoustic Hallucination | | | | | | | SA-Confusion | | | Mean |
|---|---|---|---|---|---|---|---|---|---|---|---|---|---|---|---|---|
| | Homo. | Poly. | Proso. | Knowl. | Instr. | SrcNum. | Exist. | Dist. | Dur. | Temp. | Repet. | Auth. | Inf$_a$. | Inf$_s$. | Overrel. | |
| Random | 51.25 | 49.88 | 47.36 | 49.22 | 0.00 | 53.12 | 50.00 | 48.61 | 51.39 | 52.12 | 50.52 | 50.42 | 42.08 | 48.75 | 51.56 | 49.95 |
| *Open-Source LALMs* | | | | | | | | | | | | | | | | |
| GLM4-Voice | 77.14 | 52.43 | 52.64 | 59.38 | 0.00 | 45.94 | 47.92 | 52.08 | 54.06 | 49.75 | 40.00 | 48.23 | 50.00 | **54.17** | 49.61 | 51.57 |
| SALMONN | 56.25 | 44.44 | 59.38 | 64.26 | 0.00 | 50.00 | 58.18 | 54.17 | 50.00 | 46.50 | 50.00 | 50.00 | 51.79 | 50.00 | 53.12 | 51.02 |
| Qwen-Audio | 62.32 | 57.64 | 52.40 | 57.62 | 30.00 | 46.56 | 53.27 | 47.92 | 48.13 | 62.37 | 58.44 | 50.52 | 45.98 | 52.92 | 48.05 | 53.86 |
| Qwen2-Audio | 59.46 | 43.98 | 55.29 | 67.19 | 45.00 | 50.00 | **75.15** | 47.14 | 38.75 | 48.13 | 51.56 | 50.21 | 54.91 | 50.83 | 53.12 | 53.31 |
| Qwen2-Audio-Inst | 61.07 | 46.64 | 51.44 | 73.63 | 75.00 | 50.00 | 65.33 | 58.59 | 52.81 | 49.50 | 57.50 | 58.65 | 52.23 | 39.79 | 53.12 | 55.68 |
| Kimi-Audio | 72.32 | 46.53 | 52.40 | 59.18 | **90.00** | 50.94 | 73.36 | 58.33 | 42.50 | 56.50 | 54.37 | 52.81 | 52.68 | 48.96 | **68.36** | 57.24 |
| *Closed-Source LALMs* | | | | | | | | | | | | | | | | |
| Gemini-2.5-Pro | **80.00** | **75.93** | **65.38** | **90.62** | 60.00 | **65.00** | 73.81 | **62.50** | **75.00** | **78.00** | **67.50** | **69.17** | **89.29** | 46.67 | 56.25 | **71.63** |

Table 9: Language hallucination test on AHa-Bench.

| Model | Semantic | | | Acoustic | | Mean |
|---|---|---|---|---|---|---|
| | Homo. | Klg. | Pros. | Auth. | Exist. | |
| Open-Source LALMs. | | | | | | |
| GLM4-Voice | **0.28** | 0.86 | 0.59 | 0.88 | 0.45 | 0.61 |
| SALMONN-13B | 0.61 | 0.55 | 0.86 | 1.00 | 0.64 | 0.73 |
| Qwen-Audio | 0.33 | 0.31 | **0.45** | 0.34 | **0.00** | **0.29** |
| Qwen2-Audio | 0.61 | 0.34 | 0.68 | **0.28** | 0.09 | 0.40 |
| Qwen2-Audio-Inst | 0.50 | 0.41 | 0.82 | 0.74 | 0.36 | 0.57 |
| Kimi-Audio | 0.39 | 0.69 | 0.68 | 0.81 | 0.18 | 0.55 |
| Closed-Source LALMs. | | | | | | |
| Gemini-2.5-Pro | 0.50 | **0.17** | 0.68 | 0.52 | 0.36 | 0.45 |

# E   Failure Cases

To provide researchers with a deeper understanding of how existing Large Audio-Language Models (LALMs) handle various types of audio hallucinations, we present some failure cases here across different hallucination categories for each model. The cases are also shown in our demo page at https://aha-bench.github.io/.

- Semantic Hallucination
    - Homophone Hallucination: Figure 5
    - Polysemy Hallucination: Figure 6
    - Prosodic Hallucination: Figure 7
    - Knowledge Hallucination: Figure 8
    - Instruction Hallucination: Figure 9
- Acoustic Hallucination
    - Existence Hallucination: Figure 10
    - Source Number Hallucination: Figure 11
    - Duration Hallucination: Figure 12
    - Distance Hallucination: Figure 13
    - Temporal Hallucination: Figure 14
    - Repetition Hallucination: Figure 15
    - Authenticity Hallucination: Figure 16

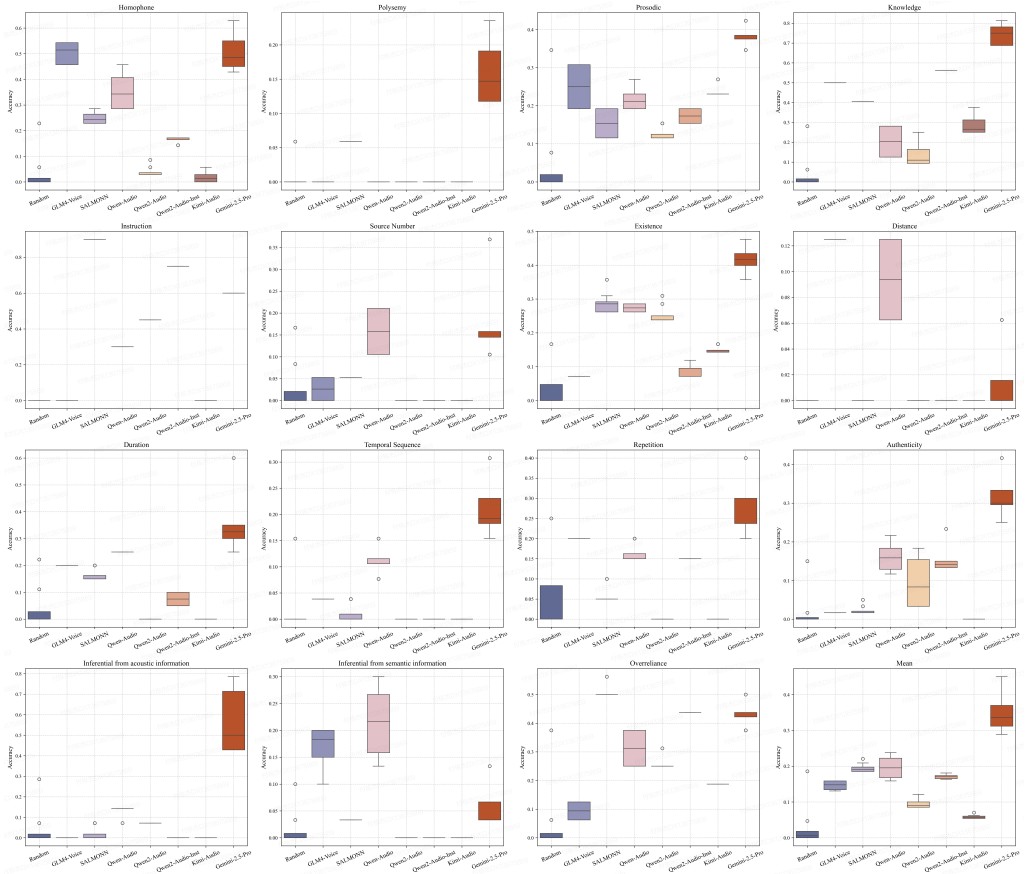

Figure 4: Error Analysis of LALMs Across Different Hallucination Types on AHa-Bench.

- Semantic-Acoustic Hallucination
  - Inferential Hallucination: Figure 17
  - Over-Reliance Hallucination: Figure 18

# F   Licenses for existing assets

- CosyVoice 1.0 [11]: Apache License 2.0
- AudioSet [17]: CC-BY-4.0
- VocalSketch [5]: CC-BY-4.0
- Kimi-Audio Evalkit [10]: MIT License
- AHa-Bench (ours): CC-BY-4.0

# G   Limitation

Due to the difficulty in collecting certain types of hallucination samples (e.g., prosodic and distance hallucinations), the sample size of our dataset is relatively modest, comparable to similar datasets, which somewhat limits its generalization capability. For instance, we are unable to comprehensively assess the model's understanding of distance variations across diverse sound-emitting objects. Nevertheless, during data selection, we endeavored to balance category distributions to achieve more comprehensive evaluation coverage. Additionally, we employed multiple sampling strategies to enhance the robustness of our benchmark.

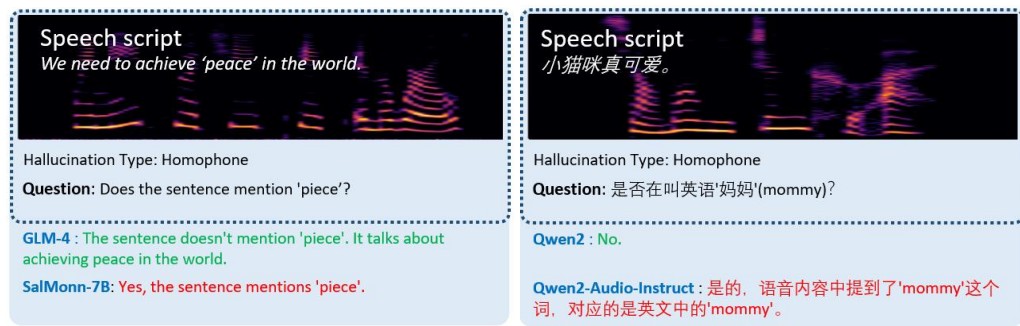

Figure 5: Failure Cases on Homophone Hallucination. Return to the Failure Case List (Section E).

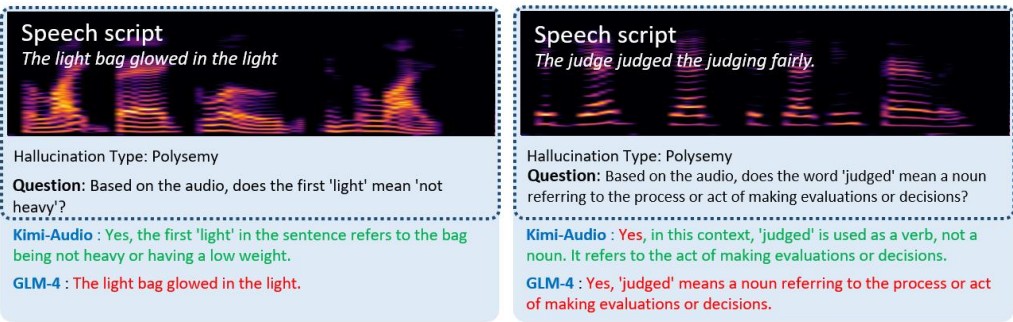

Figure 6: Failure Cases on Polysemy Hallucination. Return to the Failure Case List (Section E).

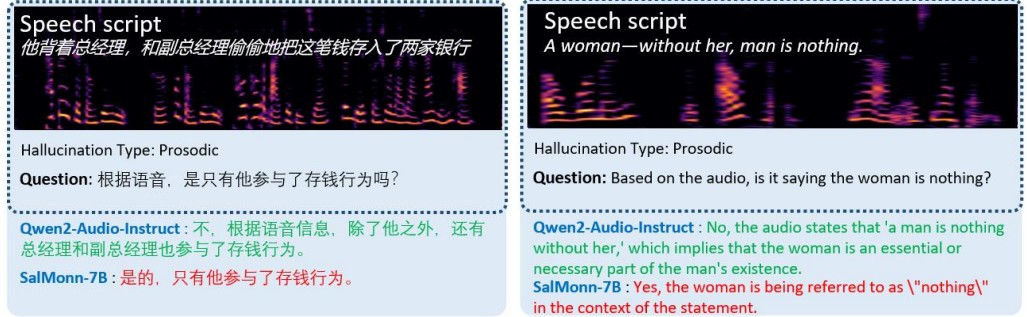

Figure 7: Failure Cases on Prosodic Hallucination. Return to the Failure Case List (Section E).

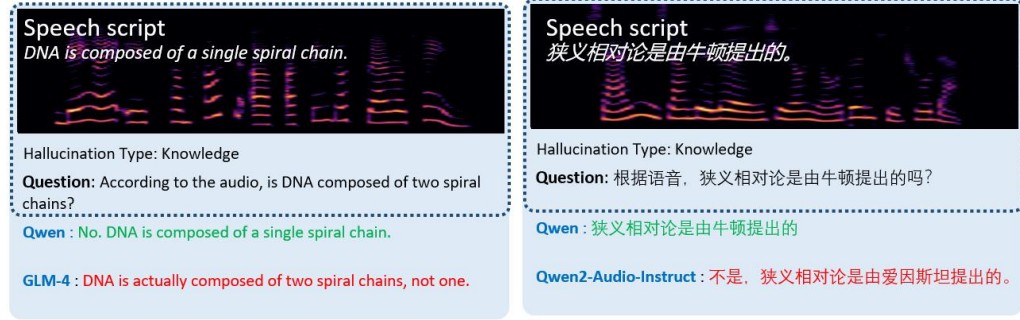

Figure 8: Failure Cases on Knowledge Hallucination. Return to the Failure Case List (Section E).

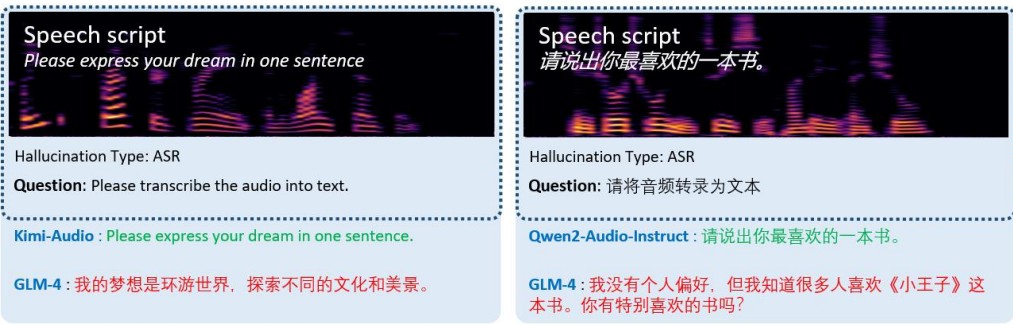

Figure 9: Failure Cases on Instruction Hallucination. Return to the Failure Case List (Section E).

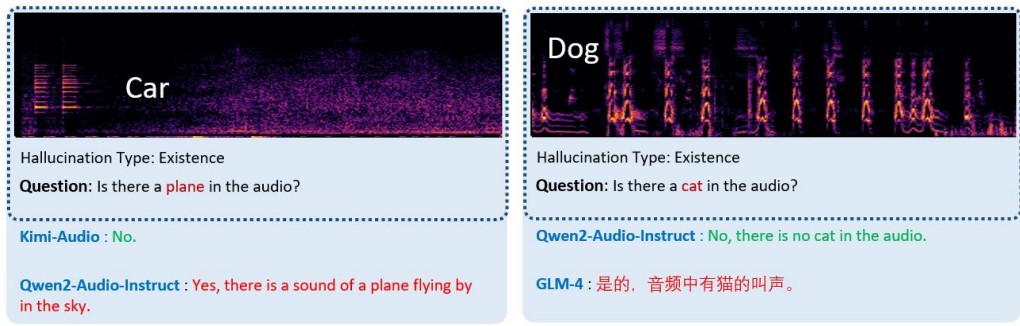

Figure 10: Failure Cases on Existence Hallucination. Return to the Failure Case List (Section E).

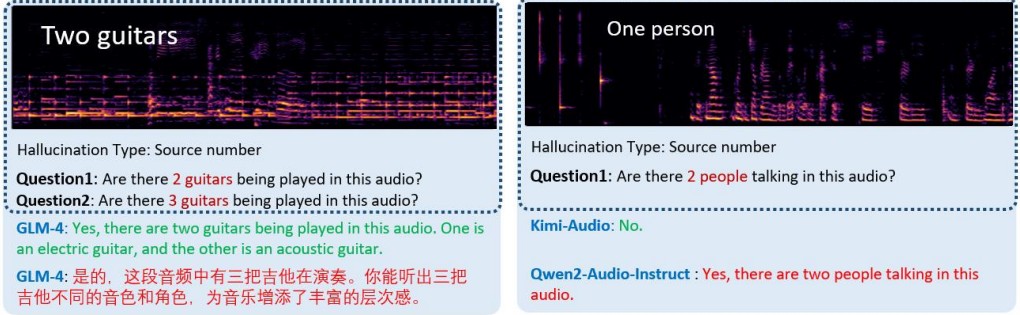

Figure 11: Failure Cases on Source Number Hallucination. Return to the Failure Case List (Section E).

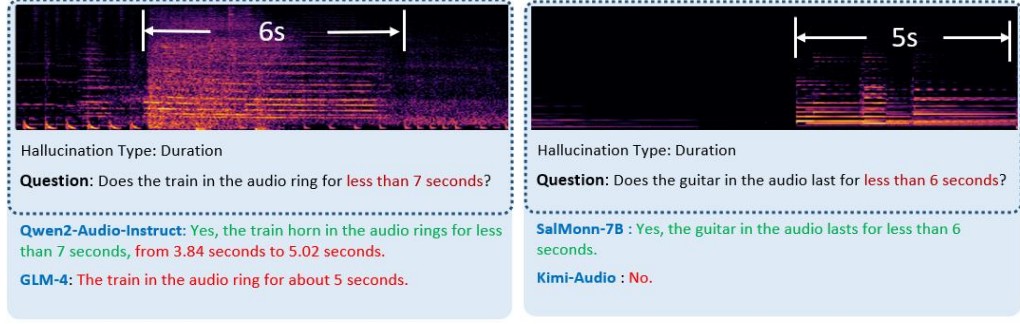

Figure 12: Failure Cases on Duration Hallucination. Return to the Failure Case List (Section E).

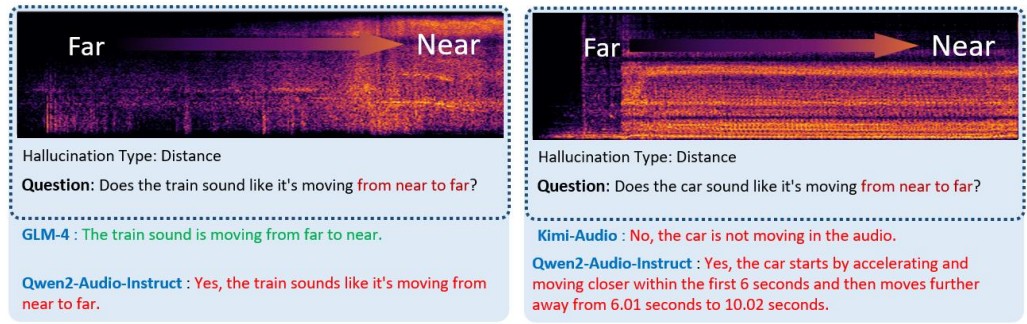

Figure 13: Failure Cases on Distance Hallucination. Return to the Failure Case List (Section E).

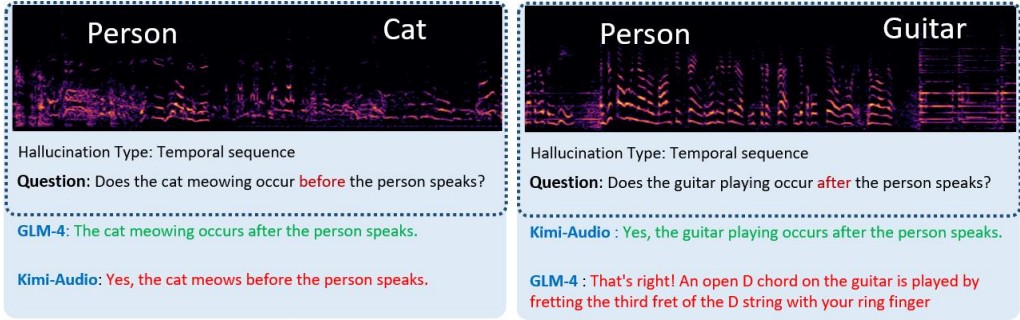

Figure 14: Failure Cases on Temporal Hallucination. Return to the Failure Case List (Section E).

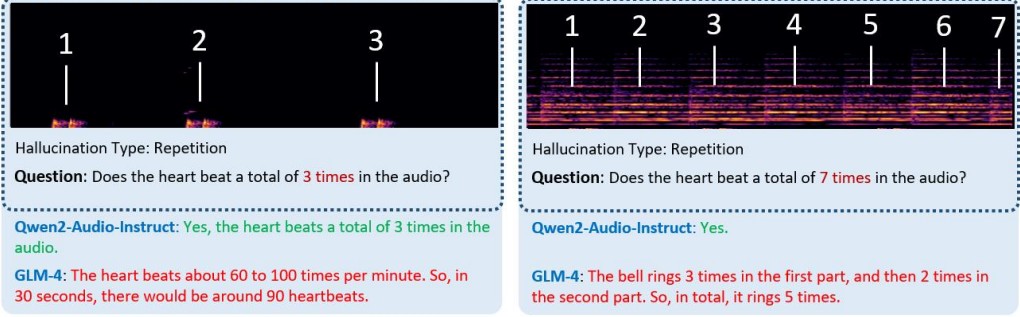

Figure 15: Failure Cases on Repetition Hallucination. Return to the Failure Case List (Section E).

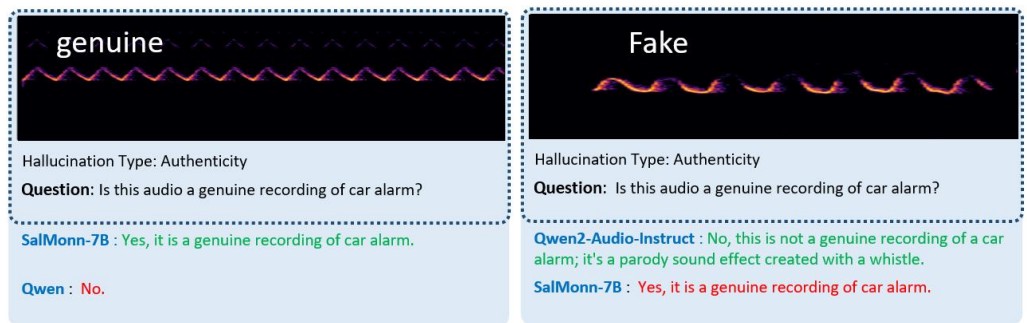

Figure 16: Failure Cases on Authenticity Hallucination. Return to the Failure Case List (Section E).

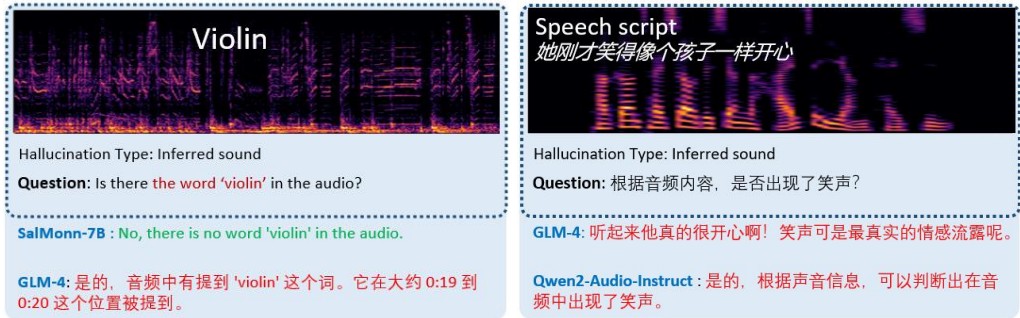

Figure 17: Failure Cases on Authenticity Hallucination. Return to the Failure Case List (Section E).

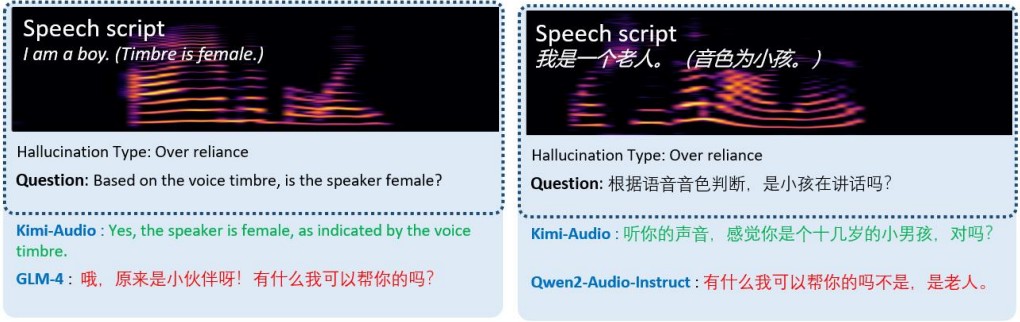

Figure 18: Failure Cases on Overreliance Hallucination. Return to the Failure Case List (Section E).

# H  Broader Impacts

Audio hallucinations pose a significant threat to the reliability of Large Audio-Language Models (LALMs), leading to potential misinterpretations of non-existent or ambiguous audio content. This can undermine the effectiveness of spoken dialogue systems, audio understanding frameworks, and intelligent customer service platforms, especially in high-stakes applications such as emergency response or assistive technologies. By introducing a comprehensive audio hallucination benchmark, this work aims to systematically evaluate and mitigate such hallucinations, promoting the development of more robust and trustworthy LALMs. We believe that this benchmark will contribute to enhancing the reliability and fairness of audio-driven AI systems, ultimately advancing the robustness of multimodal communication systems across diverse acoustic environments.

