# OpenReview forum: "AHa-Bench: Benchmarking Audio Hallucinations in Large Audio-Language Models"
_NeurIPS.cc/2025/Datasets_and_Benchmarks_Track — NeurIPS 2025 Datasets and Benchmarks Track poster_

### Official Review · Reviewer_MSYE · 2025-06-10

**Rating:** 5
**Confidence:** 3

**Summary:**

AHa-Bench is a new benchmark to evaluate auditory hallucinations in Large Audio-Language Models (LALMs). This fills a gap in research, as similar benchmarks exist for text and vision, but not comprehensively for audio. The benchmark categorizes auditory hallucinations into semantic, acoustic, and semantic-acoustic confusion types, with 14 distinct categories. It includes 396 audio samples and 906 human-annotated question-answer pairs. Initial evaluations of seven open-source LALMs demonstrate significant challenges in their joint understanding of semantic and acoustic information, highlighting critical limitations in current models.

**Dataset Code Accessibility:**

Yes

**Ethical Considerations:**

No, there are no or only very minor ethics concerns

**Limitations Weaknesses:**

A significant portion of the semantic-related hallucination audios are "synthesize[d] highly natural and realistic speech samples" using a TTS model. While verified by experts, synthesized speech might not fully capture the variability and nuances of natural human speech, potentially affecting the ecological validity of some semantic hallucination types.

The "modest" sample size, particularly for prosodic and distance hallucinations, "somewhat limits its generalization capability". It is part of the limitation of this benchmark that authors acknowledged.

The evaluation suite utilizes GPT-4o to preprocess LALM answers, categorizing them into 'Yes', 'No', and 'Unknown'. While this addresses diversity in responses, it introduces another large language model as an intermediary in the evaluation process, which could inadvertently introduce its own biases or misinterpretations.

**Strengths Contributions:**

AHa-Bench uniquely fills a critical research gap by being the first comprehensive benchmark to evaluate auditory hallucinations in Large Audio-Language Models (LALMs).

It provides a systematic taxonomy with 14 distinct hallucination types categorized into semantic, acoustic, and semantic-acoustic confusion, supported by a meticulously curated dataset of 396 audio samples and 906 human-annotated QA pairs.

The benchmark effectively reveals significant challenges in current LALMs' joint understanding of semantic and acoustic audio information, thereby setting a crucial foundation for future research in mitigating these hallucinations.

---

> ### Author Rebuttal · Authors · 2025-07-31
>
> Thank you for your recognition of our work. Please find our detailed responses to your comments below:
>
> ---
>
> - **Q1: Synthetic Speech vs. Natural Human Speech**
>
> Thank you for raising this important point. We agree that the differences between synthetic and natural human speech may introduce some evaluation bias. Nevertheless, synthetic speech plays a crucial role in simulating diverse real-world conditions and has been widely adopted in numerous spoken language understanding benchmarks [1,2].
>
> In light of your comments—as well as those from Reviewer S3UR—we have incorporated a subset of natural human speech samples to further assess audio hallucination performance. These additions will be included in the final version as a closed-source test set. Preliminary results on this extended evaluation set are shown below:
>
> | Model            | Homo  | Poly  | Proso | Knowl. |
> | ---------------- | ----- | ----- | ----- | ------ |
> | random           | 25.12 | 6.32  | 24.39 | 23.91  |
> | GLM4-Voice       | 62.18 | 0.00  | 32.12 | 50.00  |
> | SALMONN          | 12.98 | 0.00  | 23.29 | 37.50  |
> | Qwen-Audio       | 43.91 | 1.21  | 24.28 | 28.23  |
> | Qwen2-Audio      | 29.47 | 0.00  | 19.84 | 42.12  |
> | Qwen2-Audio-Inst | 21.82 | 0.00  | 18.28 | 59.89  |
> | Kimi-Audio       | 52.39 | 6.23  | 34.21 | 46.78  |
> | Gemini-2.5-pro   | 67.01 | 26.12 | 48.57 | 84.21  |
>
> ---
>
> - **Q2: Sample Size of AHA-Bench**
>
> Yes, the current scale of AHA-Bench may not extremely large. However, we believe the benchmark is sufficiently large to enable meaningful comparisons across models with respect to hallucination susceptibility. The current size is also consistent with other well-established hallucination evaluation datasets [3,4].
>
> All benchmark data have been carefully curated and manually verified. For this type of task, we believe that data quality is more crucial than quantity, and our design reflects this principle.
>
> ---
>
> - **Q3. Use of GPT-Based Metrics for Evaluation**
>
> Yes, we used GPT to assist in verifying model responses. The evaluation task was deliberately designed to be simple—limited to classifying outputs as “yes”, “no”, or “unknown”—to maximize judgment accuracy and minimize potential model bias. This approach aligns with common practice in many recent benchmarks.
>
> Nevertheless, your concern regarding GPT-based bias is valid and appreciated. To address this, we conducted an **additional human evaluation** for cross-validation. Human raters manually scored responses to assess alignment with GPT evaluations. The results, shown below, demonstrate strong consistency across metrics. These results suggest that GPT-based judgments are generally reliable for this evaluation setup.
>
> | Evaluation Source      | Homo  | Poly  | Proso | Knowl. | Instr. | SrcNum. | Exist. | Dist. | Dur.  | Temp. | Repet. | Auth. | Infa  | Infs  | Overrel |
> | ---------------------- | ----- | ----- | ----- | ------ | ------ | ------- | ------ | ----- | ----- | ----- | ------ | ----- | ----- | ----- | ------- |
> | Gemini-2.5-pro (Human) | 64.18 | 24.79 | 42.12 | 82.00  | 60.00  | 37.34   | 49.76  | 6.25  | 62.50 | 31.39 | 41.87  | 41.43 | 76.62 | 14.20 | 52.38   |
> | Gemini-2.5-pro (GPT)   | 62.86 | 23.53 | 42.31 | 81.25  | 60.00  | 36.84   | 47.62  | 6.25  | 60.00 | 30.77 | 40.00  | 41.67 | 78.57 | 13.33 | 50.00   |
>
> Thank you once again for your thoughtful feedback and valuable suggestions. Should you have any further questions or ideas, we would be delighted to continue the discussion.
>
> ---
>
> **References**
>
> [1] VoxDialogue: Can Spoken Dialogue Systems Understand Information Beyond Words? (ICLR 2025)
>
> [2] LLAMA-OMNI: Seamless Speech Interaction with Large Language Models (ICLR 2025)
>
> [3] HALLUSIONBENCH: An Advanced Diagnostic Suite for Entangled Language Hallucination and Visual Illusion in Large Vision-Language Models (CVPR 2025)
>
> [4] Mitigating hallucination in large multi-modal models via robust instruction tuning. (CVPR 2025)

---

> ### Comment · Reviewer_MSYE · 2025-08-05
>
> I would like to thank authors for their thoughtful rebuttal. Most of my concerns have been addressed, and I would like to keep my original rating as Accept.

---

### Official Review · Reviewer_Xkvg · 2025-06-20

**Rating:** 5
**Confidence:** 3

**Summary:**

This work introduces AHa-Bench, a systematic and comprehensive benchmark for audio hallucinations. AHa-Bench categorizes  these hallucinations into semantic hallucinations, acoustic hallucinations, and semantic-acoustic confusion hallucinations.  This work evaluates seven open-source local perception language models (LALMs), demonstrating the challenges these models face in audio understanding.

**Dataset Code Accessibility:**

Yes

**Dataset Code Comments:**

The dataset is available on the HF platform, which is simple and easy to access.

**Ethical Considerations:**

No, there are no or only very minor ethics concerns

**Limitations Weaknesses:**

1. The overall amount of data is not very large, and the authors, for semantic-related hallucinations, used TTS to generate the samples, a method that could theoretically produce a lot of data, what made the authors ultimately decide to include only 906 QA pairs.
2. Why is there a big performance difference between Qwen2-Audio and Qwen-Audio, when the performance of the Qwen2 model would be much better?

**Strengths Contributions:**

1. The paper is well structured and organized with a great story line.
2. The definition of Audio Hallucination is clear with the categorization, and the figures are very good.

---

> ### Author Rebuttal · Authors · 2025-07-31
>
> Thank you for your recognition of our paper’s writing and task formulation. Please find our detailed responses to your comments below:
>
> ---
>
> - **Q1: The scale of AHA-Bench**
>
> Thank you for your question. While it is true that we leverage text-to-speech (TTS) technology to reduce the difficulty of generating speech samples, the construction of the dataset still involves many manual steps that limit its scalability.
>
> Specifically:
>
> 1. **All question prompts need to be manually written**. Not every prompt is suitable for inducing hallucinations such as prosodic or homophone errors. We must carefully design the accompanying speech content to elicit specific types of hallucinations.
>
> 2. **TTS outputs that are likely to trigger hallucinations require meticulous human verification**. For example, in cases involving polysemy (e.g., Chinese characters with multiple pronunciations) or prosodic hallucinations (e.g., stress or intonation variations), we place particular emphasis on the accuracy of phoneme realization and rhythm control. However, off-the-shelf TTS systems may produce results that deviate from our intended design in these difficult cases, so manual inspection and correction are essential.
>
> These additional efforts—aimed at achieving precise alignment between linguistic input and acoustic realization—naturally constrain the overall scalability of the dataset.
>
> ---
>
> - **Q2: Why Does Qwen-Audio Perform Better Than Qwen2-Audio?**
>
> This is a key reason for developing our benchmark: models that perform well overall are not necessarily immune to hallucinations.
>
> As noted in Line 279, although Qwen2-Audio generally exhibits stronger overall capabilities, it also demonstrates greater response flexibility, which in turn leads to a more pronounced affirmative bias (i.e., a tendency to answer “yes”). This makes the model more prone to hallucinations, as shown in Table 3.
>
> In contrast, Qwen-Audio adopts a more conservative answering strategy, resulting in responses that are less expressive but more consistent. As a result, it achieves better performance under hallucination-sensitive conditions.
>
> Our benchmark is specifically designed to **identify and quantify model tendencies such as sycophancy**, which can be a major source of hallucinations. We hope this will contribute to future efforts to build more reliable and grounded multi-modal language models.
>
> ---
>
> Thank you once again for your thoughtful comments and suggestions. Please feel free to reach out with any further questions or feedback—we look forward to continued discussions!

---

### Official Review · Reviewer_iN9r · 2025-06-30

**Rating:** 5
**Confidence:** 3

**Summary:**

The paper presents a dataset for benchmarking the robustness of multimodal audio-language models to a number of failure modes ('hallucinations') due to confusion both within and between modalities. A framework for classifying the different types of hallucinations is presented, dividing hallucinations into three high-level categories - semantic, semantic-acoustic, and acoustic - and 14 sub-categories.

Audio examples are generated using text to speech (TTS) models for speech, with non-speech audio sampled from the test sets of three existing datasets: AudioSet, Compa and VocalSketch. A total of 396 audio samples are provided, along with 906 human annotated question-answer pairs, with yes-no answers, each associated with an audio sample.

Seven large audio language models (LALMs) are evaluated using the dataset, with GPT-4o used to distil the response from each model into a final 'yes', 'no' or 'unsure' answer.

A number of metrics are reported for each model, including accuracy, yes-no bias, and logical consistency between responses.

**Dataset Code Accessibility:**

Partly

**Dataset Code Comments:**

The dataset is hosted on Huggingface and appears on basic inspection to be complete and functional.
No licence for the dataset is provided on Huggingface. This should be addressed.

The fact that the dataset is, in part, derived from existing datasets should be explicitly noted in the metadata on Huggingface, and the authors should convince themselves that the licence that they place the dataset under is compatible with the licences for the underlying source datasets.

Code for evaluation of models is provided, hosted on GitHub. A readme file is present and appears to be comprehensive enough to run the model evaluation.
Again no licence is provided with the code on GitHub. This should be addressed before publication.

It is not clear whether the weights for the various models being evaluated are automatically downloaded by the script to evaluate model performance, or whether these need downloading manually. I suspect the latter based on cursory inspection of the code. If this is the case, instructions should be provided on how to do this.

**Ethical Considerations:**

No, there are no or only very minor ethics concerns

**Final Justification:**

The authors have addressed my concerns during the review process. I am happy to maintain my original rating.

**Limitations Weaknesses:**

The GPT-4o based system (section 4.2) for distilling a model under test's responses into a yes / no / unknown response is a cause for some minor concern since this might be a source of systematic bias in the performance of the evaluation. Did the authors undertake any validation of the performance of GTP-4o in converting answers in this way? I would recommend adding an analysis of the performance of this system compared with human experts, and an analysis of the performance on the outputs of the various models under test.

**Strengths Contributions:**

The paper provides a novel framework for considering the various classes of audio hallucination exhibited by large audio language models, and a dataset for their evaluation. This constitutes a valuable contribution to an emerging field of principled analysis of the failure modes of multimodal models.

The evaluation across multiple different models, with a range of informative metrics appears to be sound and useful. The framework and evaluation criteria are carefully designed, and the efforts to create a well-balanced dataset in terms of yes vs no responses is particularly useful in order to test the yes/no bias of individual models.

I am not familiar enough with the existing literature in this area to comment with certainty on the literature review portion of the paper, but the cited works appear to be relevant, and the delta that this paper provides over the existing work is clearly articulated in Table 1.
The paper is well-presented, with a clear structure. The discussion of the metrics, and their evaluation, is clear.

---

> ### Author Rebuttal · Authors · 2025-07-31
>
> Thank you for acknowledging the validity and value of our work. Please allow us to address your question in detail:
>
> ---
>
> - **Q1: Use of GPT-Based Metrics for Evaluation**
>
> Yes, we employed GPT to assist with response verification. However, the evaluation task was intentionally designed to be **simple and constrained**—requiring only a classification among *“yes”*, *“no”*, or *“unknown”*. This setup minimizes ambiguity and helps reduce the influence of potential model bias. The use of GPT-based evaluation for such tasks has become standard practice in many recent benchmarks [1,2].
>
> Nevertheless, your concern regarding the objectivity of GPT evaluations is both valid and appreciated. To further verify the reliability of our results, we conducted an additional human evaluation. Each response was manually scored to assess alignment with the GPT-based metric. The comparison below demonstrates a high degree of consistency between human and GPT ratings, indicating minimal bias in automated evaluation. These results support the conclusion that GPT-based metrics are generally reliable under our evaluation framework and task design.
>
>
> | Evaluation Source      | Homo  | Poly  | Proso | Knowl. | Instr. | SrcNum. | Exist. | Dist. | Dur.  | Temp. | Repet. | Auth. | Infa  | Infs  | Overrel |
> | ---------------------- | ----- | ----- | ----- | ------ | ------ | ------- | ------ | ----- | ----- | ----- | ------ | ----- | ----- | ----- | ------- |
> | Gemini-2.5-pro (Human) | 64.18 | 24.79 | 42.12 | 82.00  | 60.00  | 37.34   | 49.76  | 6.25  | 62.50 | 31.39 | 41.87  | 41.43 | 76.62 | 14.20 | 52.38   |
> | Gemini-2.5-pro (GPT)   | 62.86 | 23.53 | 42.31 | 81.25  | 60.00  | 36.84   | 47.62  | 6.25  | 60.00 | 30.77 | 40.00  | 41.67 | 78.57 | 13.33 | 50.00   |
>
>
>
> Thank you once again for your thoughtful feedback and valuable suggestions. Should you have any further questions or ideas, we would be delighted to continue the discussion.
>
> ---
>
> **References**
>
>
> [1] HALLUSIONBENCH: An Advanced Diagnostic Suite for Entangled Language Hallucination and Visual Illusion in Large Vision-Language Models (CVPR 2025)
>
> [2] MM-Vet: Evaluating Large Multimodal Models for Integrated Capabilities. (ICML 2024)

---

> > ### Comment · Reviewer_iN9r · 2025-08-01
> >
> > Thank you for thoroughly addressing the concerns around the objectivity of the GPT based evaluation which I see were also raised by other reviewers. The results you have provided to that end are clear and unambiguous.
> >
> > My only remaining concerns are those which I raised in the 'Datasets and Code' section around clarifying the licence for the dataset and code. I trust that these will be addressed.

---

> > > ### Author Response · Authors · 2025-08-05
> > >
> > > Thank you for your response. All our data and code will be released under the CC BY-NC 4.0 license, with appropriate attribution provided. We are also considering integrating the code into evaluation toolkits such as Kimi-Audio-Evalkit [1] to allow users to easily load different model checkpoints for streamlined evaluation. Thank you again for your valuable suggestions on our work!
> > >
> > > [1] Kimi-Audio Technical Report.

---

### Official Review · Reviewer_S3UR · 2025-07-02

**Rating:** 5
**Confidence:** 2

**Summary:**

The work creates a benchmark to investigate the performance of audio-based models in the presence of what is defined in the large language community as "Hallucinations". This includes cases where the model either under-performs due to failed understanding of the input or due to "faithfulness hallucinations" in which the model does not perform according to the intended behavior.
The authors define an ontology of different hallucination types. Then, they hand-craft a dataset where samples are designed to fall into one of several hallucination categories; the samples should lead the model to hallucinate an output that falls into the respective categories.
The authors then test several commonly used models using the benchmark and investigate them further in terms of having a bias to always answer positive, understand the input and answer logically consistent.

**Additional Feedback:**

I have chosen to reduce my confidence level because I do not agree with the definition of hallucinations which recently came up in the LLM literature. In my opinion those hallucinations are either hard examples which may lead to misclassification or the model gives wrong outputs even though the input space is sufficiently separated which would hint at a collapsed embedding or, without going into the reasons, bad performance. For models which misclassify and underperform there are plenty of metrics, datasets and tools available since decades. There is no need to re-cast general machine learning problems as hallucinations. So in my opinion this dataset is a hard example dataset and the word hallucination is lumping together separate issues. However, that is not the author's fault. Related work defines hallucinations and the authors do their best to create an ontology for those definitions. Therefore I judge this work under the assumption that hallucinations are a valid framework to discuss these effects even though in general I have a different opinion.

**Dataset Code Accessibility:**

Yes

**Dataset Code Comments:**

The dataset is very well shared and the Github repository provides very clear instructions and a good method to get and test the benchmark. There is extra code provided to download the data from Huggingface which converted the data to a harder-to-handle parquet files.

**Ethical Comments:**

This concern does not fit in any of the "Ethics Flags" categories. By publishing all samples of the benchmark, model creators can finetune on the benchmark. So performance of new models can not be measured on the benchmark. It would be potentially wise to keep a part of the dataset undisclosed from the general public. Since the data is already published, if accepted maybe it could be good to add a small amount of extra samples for each category only to see if models perform significantly worse on that hidden category.

**Ethical Considerations:**

Yes, there are ethics concerns that require attention by the authors

**Final Justification:**

The authors have answered my questions sufficiently. The evaluation in Q1 shows the benchmark design is valid. The evaluations in Q3 show that real speech samples lead to similar results as synthesized ones, lending credibility to the method. I increased my rating to accept.

**Limitations Weaknesses:**

The GPT-based evaluation of positive and negative responses probably works but is a source of error. Statistics of yes/no/unknown outputs should be given and it would have been probably good if a human tested a number of samples, meaning a human should do a sanity check for a number of examples if GPT got the right response type.

While hand-crafting these examples is surely time consuming the dataset seems to be still relatively small. However, this is a minor point because I do not see a practical way to speed up generation of these samples or add to it without significant human work.

Another issue of hand-crafting is finding hallucination inducing instances. It should be a bit better explained how these instances were found. If finding them relies only on human knowledge that may bias the benchmark towards examples humans find confusing or hard. Potentially it would be better to look at a number of models and see where their performance is bad and then under those hard examples look for instances which fit to a hallucination category.

This dataset technically mostly tests whole categories, such as the semantic hallucination, only on text-to-speech generated samples. If there is a difference between the sensitivity to hallucinations between real samples and generated samples is unknown. It would have been potentially good to create a small set of real samples, e.g. only for speech understanding which can be realistically done, and see if the model evaluations more or less align with the generated input.

The font in Figure 4 is tiny. Please try to find ways to increase the font size.

**Strengths Contributions:**

The paper is very clearly written. Examples for hallucinations are very well visualized, e.g. in Figure 3 where the meaning of a distance hallucination is very clear and what is a wrong and what is a correct model answer is illustrated. All statistics are clearly defined and performances on the various metrics well shown. Figure captions explain very well the content.

The investigation into several models in Table 2 is interesting and the performance is roughly similar to comparable investigations into general performance of these models. This gives some confidence that the method of the authors is suitable as a proxy for general model performance.

The hand-crafted creation of these examples is probably very time and labor consuming. Given my assumption (see additional feedback) the authors have chosen a good method to systematically find hard cases, similar to how NHTSA categories were used for the Carla simulator. However, in this case there can be no easy data driven creation of these cases and so the effort of the authors is commendable.

The dataset is very well shared and the Github repository provides very clear instructions and a good method to get and test the benchmark

---

> ### Author Rebuttal · Authors · 2025-07-31
>
> We sincerely appreciate your thoughtful and constructive feedback on our work. Please find our detailed responses below.
>
> ---
>
> - **Q1: The Use of GPT-Based Metrics**
>
> Thank you for highlighting concerns regarding the use of GPT-based evaluation. Our setup aligns with recent practices [1, 2] in evaluating hallucinations and reasoning abilities in LLMs, and is increasingly regarded as an effective and reproducible solution for benchmark design. It ensures consistency and fairness in future evaluations by other researchers.
>
> That said, we fully acknowledge the potential for model-induced bias. To address this, we conducted an additional round of human evaluation to assess the reliability of the GPT-based metric. Each sample was independently annotated by human evaluators. As shown in the table below, there is strong alignment between GPT and human ratings, indicating that the GPT-based metric offers a consistent and reasonably accurate approximation for our task setting:
>
> | Evaluation Source      | Homo  | Poly  | Proso | Knowl. | Instr. | SrcNum. | Exist. | Dist. | Dur.  | Temp. | Repet. | Auth. | Infa  | Infs  | Overrel |
> | ---------------------- | ----- | ----- | ----- | ------ | ------ | ------- | ------ | ----- | ----- | ----- | ------ | ----- | ----- | ----- | ------- |
> | Gemini-2.5-pro (Human) | 64.18 | 24.79 | 42.12 | 82.00  | 60.00  | 37.34   | 49.76  | 6.25  | 62.50 | 31.39 | 41.87  | 41.43 | 76.62 | 14.20 | 52.38   |
> | Gemini-2.5-pro (GPT)   | 62.86 | 23.53 | 42.31 | 81.25  | 60.00  | 36.84   | 47.62  | 6.25  | 60.00 | 30.77 | 40.00  | 41.67 | 78.57 | 13.33 | 50.00   |
>
> ---
>
> - **Q2: Potential Bias from Human Knowledge in Annotation**
>
> We appreciate your insightful observation. Indeed, building automated, human-knowledge-independent evaluation pipelines is an exciting and promising direction for future research. However, it remains a highly challenging task. Key open questions include:
>
> 1. How to systematically identify truly challenging examples;
> 2. How to automatically generate discriminative and complex prompts;
> 3. How to verify model outputs without relying on human judgment.
>
> Such goals may require sophisticated agent-based or feedback-driven systems, which are beyond the scope of this work. For now, our goal is to present a comprehensive and operational definition of audio hallucinations, even if partially shaped by human intuition, in order to provide a foundation for future research in this space. We hope our benchmark can stimulate further work toward more objective and automated evaluation methodologies.
>
> ---
>
> - **Q3: Synthetic Speech vs. Natural Human Speech**
>
> Thank you for raising this important point. While synthetic speech provides a controlled, diverse, and scalable evaluation setting, and is widely adopted in many spoken language understanding benchmarks [3, 4], we agree that it may not fully reflect real-world characteristics of natural speech.
>
> In response to your feedback, we have incorporated a subset of real human speech samples into our evaluation. These will form part of a closed-source test set to be included in the final release. Preliminary results on this extended evaluation set are summarized as follows:
>
> | Model            | Homo  | Poly  | Proso | Knowl. |
> | ---------------- | ----- | ----- | ----- | ------ |
> | random           | 25.12 | 6.32  | 24.39 | 23.91  |
> | GLM4-Voice       | 62.18 | 0.00  | 32.12 | 50.00  |
> | SALMONN          | 12.98 | 0.00  | 23.29 | 37.50  |
> | Qwen-Audio       | 43.91 | 1.21  | 24.28 | 28.23  |
> | Qwen2-Audio      | 29.47 | 0.00  | 19.84 | 42.12  |
> | Qwen2-Audio-Inst | 21.82 | 0.00  | 18.28 | 59.89  |
> | Kimi-Audio       | 52.39 | 6.23  | 34.21 | 46.78  |
> | Gemini-2.5-pro   | 67.01 | 26.12 | 48.57 | 84.21  |
>
> ---
>
> - **Q4: Benchmark Leakage Concerns**
>
> We fully agree that making evaluation data publicly available can lead to benchmark overfitting and reduced credibility. To mitigate this risk, we are building a private, closed-source test set composed of previously unreleased samples. This set will be reserved for future model evaluations to ensure fairness, integrity, and generalizability of the benchmark.
>
> ---
>
> - **Q5: Hard Cases vs. Hallucination Cases**
>
> Thank you for your nuanced discussion of this distinction. We would like to share our view developed during the study: hallucinations are often “easy” cases where the model should succeed, but fails due to overconfidence or improper reasoning, while hard examples are inherently difficult, even for well-trained models.
>
> For example, in overreliance hallucinations, the model is expected to extract cues like speaker age from acoustic signals, but instead overrelies on irrelevant semantic information, leading to hallucination. Similarly, in instruction hallucination cases, we found that imperative-sentence prompts are more likely to trigger erroneous responses.
>
> Admittedly, some newly introduced task categories—such as distance hallucination—may lie near the boundary between hallucinations and difficult cases. We recognize this ambiguity and commit to continuously updating our hallucination test set as community models advance to clarify the distinction between hard and hallucination cases.
>
> ---
>
> Once again, thank you for your valuable suggestions and critical insights. We welcome further discussion and hope our responses have addressed your concerns thoroughly.
>
> ---
>
> **References**
>
> [1] HALLUSIONBENCH: An Advanced Diagnostic Suite for Entangled Language Hallucination and Visual Illusion in Large Vision-Language Models (CVPR 2025)
>
> [2] Mitigating hallucination in large multi-modal models via robust instruction tuning. (CVPR 2025)
>
> [3] VoxDialogue: Can Spoken Dialogue Systems Understand Information Beyond Words? (ICLR 2025)
>
> [4] LLAMA-OMNI: Seamless Speech Interaction with Large Language Models (ICLR 2025)

---

> > ### Comment · Reviewer_S3UR · 2025-08-05
> >
> > Thank you for the rebuttal, most of my concerns have been alleviated. Comparing q3 with Table 2 it seems real human speech sampled evaluations show the same ranking and similar evaluations to the synthesized data. Is that a correct insight?

---

> > > ### Author Response · Authors · 2025-08-05
> > >
> > > Thank you for acknowledging our response. Your insightful discussion on challenging and hallucinated cases has helped clarify the significance of our work. We also sincerely appreciate your decision to raise the score for our submission.
> > >
> > > Indeed, Table 2 and the results from Q3 lead to similar conclusions. Due to the limited rebuttal timeframe, the semantic content of the natural human speech and the synthesized speech remains largely consistent. The experiments in Q3 and Table 2 together demonstrate that our synthesis method can serve as a viable approach for constructing benchmark evaluation sets, capturing the general performance trends of models.
> > >
> > > Nevertheless, as you rightly pointed out, only real data can truly reflect real-world evaluation outcomes. Therefore, we plan to rely entirely on natural human speech in our closed-source test set. Furthermore, since parts of this closed test set overlap with our released data, we are actively augmenting it with more diverse samples to enhance the robustness of our evaluation results.
> > >
> > > Thank you again for your valuable suggestions on our work!

---

### Note · Authors · 2025-08-12

We sincerely thank all reviewers and the AC for their efforts in evaluating our work. At the conclusion of the rebuttal stage, please allow me to provide a final summary of the changes and clarifications we have made:

First, all reviewers expressed recognition of our **Aha-Bench** and acknowledged the significance of defining audio hallucinations. Reviewers **S3UR**, **Xkvg**, and **iN9r** appreciated the convenience brought by the open-sourcing of our work. Reviewers **S3UR** and **Xkvg** also commended the quality of our writing. Reviewers **MSYE** and **S3UR** considered our work to be of substantial importance for future research.

During the rebuttal stage:

* Following the suggestions of reviewers **S3UR** and **MSYE**, we verified the consistency of hallucination phenomena between real human recordings and synthetic data, and we committed to including more real human recordings in future work for a more realistic and robust demonstration.
* Following the suggestions of **S3UR**, **iN9r**, and **MSYE**, we validated the consistency between GPT-based metrics and human evaluations.
* Following the suggestion of **S3UR**, we committed to maintaining an additional closed-source evaluation set in the future to ensure fairness in benchmarking.
* We have also made every effort to ensure that **Aha-Bench** avoids potential ethical concerns.

Finally, we once again express our sincere gratitude to all reviewers for their contributions, constructive feedback, and positive evaluations, which have helped make **Aha-Bench** more complete and impactful.

---

### Decision · Program_Chairs · 2025-09-18

**Decision:**

Accept (poster)

**Comment:**

This submission presents AHa-Bench, a new benchmark for evaluating auditory hallucinations in Large Audio-Language Models (LALMs). The benchmark defines an ontology of hallucination types—including semantic, acoustic, and semantic-acoustic confusion—with 14 sub-categories. The dataset consists of 396 audio samples and 906 human-annotated question-answer pairs. The authors evaluate seven open-source LALMs and analyze performance metrics such as accuracy, yes/no bias, and logical consistency. Synthetic speech generated via TTS is used for many semantic-related hallucinations, complemented by a subset of natural human speech to validate ecological realism.
The paper introduces a valuable benchmark, provides clear evaluation methodology, and addresses reviewer concerns effectively. Minor limitations can be mitigated in camera-ready revisions.